# Using Artificial Intelligence (AI) to predict organizational agility

**Niusha Shafiabady** [1,2]*, **Nick Hadjinicolaou**[3], **Fareed Ud Din** [4], **Binayak Bhandari**[5], **Robert M. X. Wu**[6], **James Vakilian**[2]

1 Faculty of Science and Technology, Charles Darwin University, Haymarket, NSW, Australia, 2 Design and Creative Technology, Torrens University Australia, Sydney, NSW, Australia, 3 Global Project Management, Torrens University Australia, Adelaide, SA, Australia, 4 Faculty of Science, Agriculture, Business and Law, School of Science & Technology, The University of New England, Armidale, NSW, Australia, 5 ARC Training Centre for Automated Manufacture of Advanced Composites, The University of New South Wales, Sydney, NSW, Australia, 6 School of Engineering and Technology, Central Queensland University, Rockhampton, Queensland, Australia

* niusha.shafiabady@cdu.edu.au

**Data Availability Statement:** Sharing the data is not allowed because it potentially contains identifying information about the organisations where participants were associated to and sharing the data would violate the confidentiality agreement

## Abstract

Since the pandemic organizations have been required to build agility to manage risks, stakeholder engagement, improve capabilities and maturity levels to deliver on strategy. Not only is there a requirement to improve performance, a focus on employee engagement and increased use of technology have surfaced as important factors to remain competitive in the new world. Consideration of the strategic horizon, strategic foresight and support structures is required to manage critical factors for the formulation, execution and transformation of strategy. Strategic foresight and Artificial Intelligence modelling are ways to predict an organizations future agility and potential through modelling of attributes, characteristics, practices, support structures, maturity levels and other aspects of future change. The application of this can support the development of required new competencies, skills and capabilities, use of tools and develop a culture of adaptation to improve engagement and performance to successfully deliver on strategy. In this paper we apply an Artificial Intelligence model to predict an organizations level of future agility that can be used to proactively make changes to support improving the level of agility. We also explore the barriers and benefits of improved organizational agility. The research data was collected from 44 respondents in public and private Australian industry sectors. These research findings together with findings from previous studies identify practices and characteristics that contribute to organizational agility for success. This paper contributes to the ongoing discourse of these principles, practices, attributes and characteristics that will help overcome some of the barriers for organizations with limited resources to build a framework and culture of agility to deliver on strategy in a changing world.

## 1. Introduction

The concepts of organizational agility have become increasingly important since the pandemic as organisations require new ways to improve employee engagement, build capabilities and organizational performance to support competitiveness and deliver on their strategy. Other

signed with the participants. These restrictions were imposed by the Human Research Ethics Committee of Torrens University Australia. Contact information for data access inquiries is as follows: Human Research Ethics Committee of Torrens University Australia (Ethics@torrens.edu.au).

**Funding:** The authors received no specific funding for this work.

**Competing interests:** The authors have declared that no competing interests exist.

pressures include the increased use and merging of technology, use of Artificial Intelligence, greater investment in technology, greater customer needs and levels of customer satisfaction demanded. Market competition, legislative, political and economic pressures [1,2] have also played a role in the need for increased organizational and strategic agility. Organizational agility is defined as "the capability to quickly sense and adapt to external and internal changes to deliver relevant results in a productive and cost-effective manner" [3]. It is closely linked to the execution of an organisation's strategy. Benefits include the acceleration of organizational learning to meet the pace of rapid environmental change through flexibility in assembling resources, knowledge, processes and capabilities. Agility also enables organisations to sustain their competitive advantage by increasing responsiveness when making business decisions [4,5]. New policies, process and procedures together with changes to organizational structures, systems and management practices are also required [6], which also support improving levels of maturity in these areas.

In this research, characteristics and practices of organisational agility are examined together with organizational size and type, industry sector, support structures, maturity levels and other aspects of future change are applied into an artificial intelligence model to predict an organizations future agility[6,7]. Barriers and benefits of improved organizational agility are also explored through this data collection to support building an organizational knowledge and framework to support improving agility and associated attributes of maturity [8,9].

Artificial Intelligence (AI) is applicable to a vast variety of interdisciplinary business systems, with the main objective of improving standards, reducing the dependable product quality control methods and creating new possibilities to meet customer satisfaction while maintaining low operational costs and robust business management [10]. For example, AI has been implemented to improve supply chain management [11], smart factory-based operational efficiency and warehouse management [12]. AI has helped improve a range of other domains as well such as health and medicine, cancer care [13] among other avenues such as pandemic management [14,15]. The most effective AI methodologies are Machine Learning (ML) methods e.g., Support Vector Machine (SVM), Decision Trees (DT), and K-Nearest Neighbours (KNN), as detailed in the later sections, which have been utilised in this research article to predict organizational agility. Applications of AI and ML methods include commercial use of SVM [16–19], energy load management [20], finance and portfolio management, inventory management [21,22], sales forecasting, profit maximisation, and sales growth [23], among other domains e.g., dynamic cognitive networking approach [18], modelling and identifying frailty [24,25]. In order to maximise the potential of revenue generation, predicting Organizational Agility (OA) is an important factor. Hence this paper focuses on highlighting the use of AI to predict organizational agility together with barriers and benefits of improved organizational agility by incorporating a comprehensive data collection, prediction and analysis methodology.

A base collection of data was collected from 44 respondents in private and public Australian sectors. The following attributes of an organization and characteristics of organizational agility (1–17) were identified by PMI [3,26–30] and explored within the model together with components of maturity:

1. Flexible and adaptable, 2. Open communications, 3. Transparency in decision making, 4. Rapid decision making, 5. Decentralised decision making, 6. Open to change, 7. Self-aware and honest, 8. Customer orientated, 9. Focused on talent development, 10. Committed to Agility, 11. Empowered team members, 12. Action based, 13. Agility recognised as a team competence, 14. Catalyst Leadership, 15. Effective methods of rapid knowledge transfer, 16. Continuous learning from experience, 17. Clear guidelines for tailoring standardised processes to suit the size and type of project, 18. Effective environment scanning, 19. Appetite for risk, 20. Active Governance

Building on a previous study, further characteristics (18–20) were identified by Hadjinico-laou et al [6], which included effective environmental scanning, having an appetite for risk and active governance [6]. In addition to the organisational characteristics' successful implementation of organizational agility generally requires a number of ongoing practices. A PMI report (p. 6) [29] identifies the six foundational practices of organizational agility including:

1. Responding quickly to strategic opportunities, 2. Shortening production/review/decision cycles, 3. Eliminating organizational silos, 4. Aligning new business capabilities to strategy, 5. Integrating the voice of the customer and 6. Focusing on change management

These practices and others have been taken into consideration in this study where respondents were asked about the following 27 practices that have been found by PMI [30] that support organizational agility:

1. Focus on change management, 2. Application of iterative project management concepts to portfolio management, 3. Use of program management practices, 4. Focus on resource management, 5. Focus on risk management, 6. Focus on lean practices and value, 7. Project task simplification, 8. Quick response to strategic opportunities, 9. Shorter production/review/ decision cycles, 10. Elimination of organisational silos, 11. Use of Project Portfolio Management practices, 12. Integrates voice of the customer, 13. Use of iterative or incremental project management practices, 14. Interdisciplinary project teams, 15. Contingency planning, 16. Leverages technology, 17. Empirical (real-time) project management, 18. Matrix management, 19. Use of models, pilots and simulation, 20. Question assumptions, 21. Assessment of disruptive technological or other changes, 22. Increased environmental scanning, 23. Growth by acquisition, 24. Focus on innovation, 25. Standardisation of project management practices, 26. Progressively elaborated and active use of Project Business Cases, 27. Use of Business Cases for post implementation reviews and benefits management.

Some of these practices are more complex than others and required at the individual, team and organizational level. Consideration is also required on how an organisations strategy is formulated, the planning horizon, support structures required for execution and other strategic practices. Changes to an organizations culture rely heavily on strong leadership and ongoing commitment. While factors, characteristics and practices have been studied in the existing literature, little has been covered to apply Artificial Intelligence to predict organizational agility. The questionnaire survey explored respondent roles, respondent industry sector, organizational size, characteristics and practices of organizational agility, aspects of future change, maturity levels of practices, benefits and barriers to improve organizational agility and aspects of maturity to support organizational agility.

The objectives and primary research questions for this study were:

i.  What are the leading characteristics and foundational practices to support organisational agility?

ii.  How can Artificial Intelligence be used to predict organizational agility?

Data was analysed and an artificial intelligence model was developed and using this data the organizations agility was predicted. We begin by examining the literature that is relevant to our study. In Section 2, we describe our methods, including data, measures and research methodology. The research results and discussion are presented in Section 3 our overall conclusions are presented in Section 4.

## 2. Research methodology

The data that was collected for this research is sourced from 44 industry practitioners from differing industry sectors in Australia (with 42% in private sector organizations, 17% public

sector, 22% in services organizations, 11% Universities and 8% not for profit and others) using an online questionnaire survey with a semi-structured questionnaire. Definite (yes/no) and concrete answers were managed by the use of closed questions. On the other hand, any further information or opinions from the respondents that did not appear in the closed questions could be added to 'Others (please specify).' Most fields in the data comprised of categorical variables. Our analysis provides original evidence that the process of data analysis followed the four steps modified from Creswell and Plano [4] for channel analysis based on 1) Preparing the data for analysis, 2) Exploring the data, 3) Analysing the collected data, and 4) Representing the data analysis.

The questionnaire survey asked explored respondents for their role within the organisations, country, the number of employees within the organisation, the organisational type, respondent industry sector, whether the organisation had a PMO, organizational size, agile characteristics within organisation, agile and practices of the organizational agility, aspects of future change, level of maturity levels of practices within the organisation, changes that occurred within the last two years, external factors that changed, perceived benefits and barriers to improve organizational agility and aspects of maturity to support organizational agility. Responses are covered in the results and discussion section of the paper.

## 2.1 Data collection

There have been a diverse range of organisations types and sizes in the sample size covered in the results and discussion section 2.1.2 and 2.1.3 with challenges for organisational agility greater in larger organisations in the public sector.

Greater effort is required in larger organisations to improve organisational agility.

**2.1.1 Respondent roles.** The results are based on the primary roles of the respondents (refer Fig 1) were Project Manager (19%), Team Member (16%), Manager (13%), Executive General Manager (13%) and business trainers or lecturer roles within their organisation (19%).

**2.1.2 Industry sector.** The predominate industry sector was education (15%) followed by Professional Services, Hospitality and Tourism (13%) (refer Fig 2).

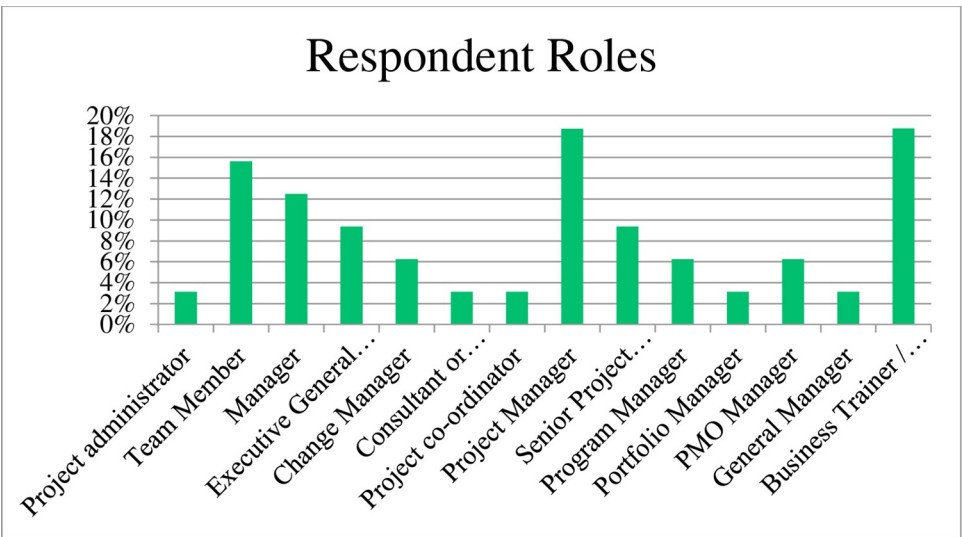

**Fig 1. Respondent roles.**

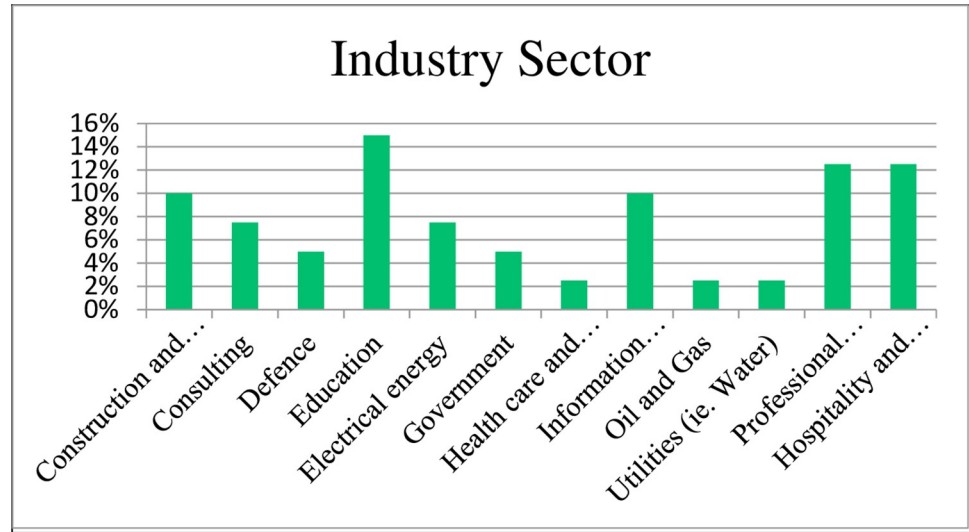

**Fig 2. Respondent industry sector.**

**2.1.3 Organisational size.** In Fig 3, a significant portion (33%) of respondents were from organisations with a size in the 300 to 2000 range with 17% less than 19 and from 100 to 299 employees (refer Fig 3).

The state-of-the-art Artificial Intelligence (AI) methods have been incorporated in this research to model and predict the organizational agility. The AI methods utilised in this paper are discussed in the following section.

## 2.2 Artificial Intelligence (AI) methods and data analysis

Artificial Intelligence has been used to solve different problems in different industries [31]. Machine learning is an evolving field of computational science that aims to imitate intelligence by incorporating learning ability and adaptability as per the environment [32]. It falls under

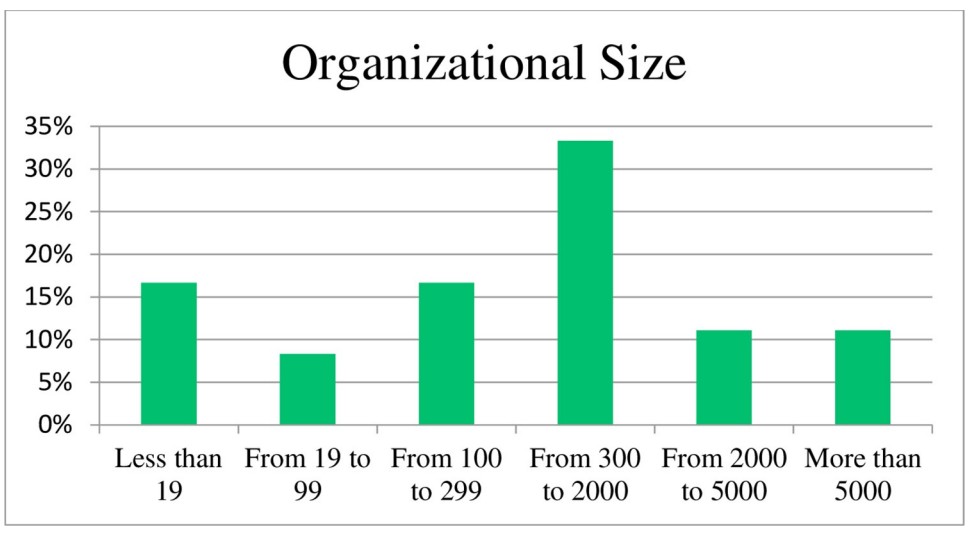

**Fig 3. Respondent organizational size.**

the umbrella of Artificial Intelligence. With the emergence of big data technologies and high-performance computing, machine learning has created new opportunities for data-intensive studies in a variety of multi-disciplinary technological disciplines [33]. Machine learning generally uses two different applied techniques: unsupervised learning and supervised learning. Unsupervised Learning utilises underlying internal structures and patterns in the data to analyse and cluster unlabelled datasets, for example, there are a number of standard clustering techniques available i.e., k-means, Gaussian mixture models, hidden Markov models, fuzzy C-means clustering, hierarchical clustering, self-organizing maps etc. In contrast, Supervised Learning uses labelled datasets to train algorithms to classify data or predict outcomes accurately e.g., via classification and regression techniques to construct machine learning models that train a model on known input and output data to predict future outcomes. There exists a range of algorithms for performing classification including Support Vector Machines (SVMs), Decision Trees (DT), k-Nearest Neighbours (KNN), Naive Bayes, Neural Networks and Logistic Regression. This paper utilises a hybrid approach comparing three effective machine learning algorithms: SVM, DT and KNN to predict organizational agility and includes managing risks, stakeholder engagement and delivering strategic outcomes, which can support the development of required new competencies, apply new skills and develop a culture of adaptation to improve performance and successfully deliver on strategy.

Data classification is a technique that is used to organize data into categories/classes that also involves tagging the data to make it easily searchable and trackable in order to make it is easy to retrieve, sort and store for future use i.e., for making predictions based of factual trends. Support Vector Machine (SVM) is a strong classification method in ML techniques that incorporate maximisation i.e., (support) of the separating margin (i.e., vector) and is commonly applied in classification and nonlinear function estimation [34]. SVM has a vast applicability in research related to text and data mining, e.g., in order to identify effective marketing tactics, where very little client information is usually superseded with the preferences of a broader population. Similarly, credit rating analysis and power price forecasts are among many other corporate applications where SVM have been used as a successful methodology [35]. SVM Hyperplane generates a decision border that distinguishes between the two data classes. Depending on which side of the hyperplane a data point falls, it links to that respective class. The goal of SVM Hyperplane algorithms for binary classification problems is to locate a hyperplane for each data class that is both distant from and proximate to the data points of the other class. SVM hyperplane algorithms perform better and have lesser computing cost when compared to the traditional SVM. However, when the dataset includes considerable noise, SVM Hyperplane does not perform very well, which calls for a hybrid approach in ML algorithms to make accurate predictions [36].

Similarly, just like SVM, k-Nearest Neighbours (KNN) is a simple but effective method for classification, which also has successfully been implemented in real-time applications. The scalability of the KNN methods to large-scale datasets makes it more adaptable for a vast range of applications [37]. The KNN has been applied and several different domains including the prediction of financial straits in commercial entities, with the goal of limiting societal losses. In some research problems, combining-classifiers system provides high performance, and KNN along with SVM has been used in recent research literature as an outperforming combining-classifiers system, especially for predicting business failures [24].

Other than SVM and KNN, another outperforming classification method is Decision Tree (DT) [38], which is commonly utilised to design prediction algorithms for a target variable or to create classification systems based on multiple covariates. By splitting the large datasets into training and validation datasets, DT can easily deal with huge, intricate datasets without imposing a sophisticated parametric structure, which makes it an effective classification technique [37].

Decision Tree has proven to be a successful methodology, especially when it comes to deciding whether or not to invest in business i.e., venture capital firms, who most of time have limitation of available information and time. Decision tree algorithm provides detailed analysis which can assist investors while performing initial screening for the suitable projects [35].

A classical machine learning approach for data classification is Naïve Bayes [39], which is usually considered the preferred methodology for binary or categorical data. Naïve Bayes is a simple machine learning methodology and doesn't require huge amounts of training data and can handle continuous as well as discrete data. A range of different studies have utilised Naïve Bayes methodology to make real-time predictions as it is highly scalable with the number of predictors and data points [32,40]. However, some recent studies have highlighted the limitations of Naïve Bayes, especially in cases that there are insufficient occurrences of some data class labels where certain attributes have similar values [39].

Another machine learning method is called gradient boosting machine (GBM), which is used for data classification and for regression problems. GBM [41] is based on the hunch of prior models coupled with the best feasible training data model to minimise the overall prediction error. Some of the benefits of GBM is being faster in training speed and providing higher accuracy. GBM has some limitations as well—for example, it will continue improving to minimize all errors, which makes it computationally expensive, time-consuming and memory exhaustive [41].

One of the frequently used supervised machine learning algorithms is called random forest (RF), which is used for data classification. On various samples, RF constructs decision trees and uses their average for classification and majority vote for regression [42]. RF offers quite a good accuracy among the categorization machine learning techniques. One of its advantages is its ability to handle large amounts of data with hundreds of different variables, with automatic balancing mechanisms where there are imbalanced classes within the dataset [31].

Logistic Regression [31,43] is another machine learning technique that is widely used to conduct supervised learning to predict different phenomena. Logistic Regression is easier to set up and train than other ML techniques. Several studies have suggested implementing Logistic Regression because of its efficiency especially when different data outcomes exist in a linearly separable manner. Some recent studies outline issues of Logistic Regression [31,43], especially when it fails to predict a continuous outcome, and its tendency to assume linearity between different variables. In some cases, when the data sample is small, the results generated by Logistic Regression may not be accurate [32,40].

Considering the benefits and applicability of these ML algorithms, in this paper, we have used an analytical method to generate synthetic data from the actual responses provided by the survey's respondents.

## 3. Results and discussion

### 3.1 Characteristics of organisational agility

The ability to adapt quickly in response to changes to sustain competitive advantage is a critical factor for success [3]. Organizational agility (OA) is the ability to adapt and accommodate changes required to respond with the required actions and requires the development of a number of characteristics and practices.

The following characteristics of organizational agility (1–17) were identified by PMI [3] and explored within this study:

1. Flexible and adaptable, 2. Open communications, 3. Transparency in decision making, 4. Rapid decision making, 5. Decentralised decision making, 6. Open to change, 7. Self-aware and honest, 8. Customer orientated, 9. Focused on talent development, 10. Committed to Agility, 11. Empowered team members, 12. Action based, 13. Agility recognised as a team

competence, 14. Catalyst Leadership, 15. Effective methods of rapid knowledge transfer, 16. Continuous learning from experience, 17. Clear guidelines for tailoring standardised processes to suit the size and type of project, 18. Effective environment scanning, 19. Appetite for risk, 20. Active Governance

Further characteristics (18–20) were identified by Hadjinicolaou et al [6] which included having an appetite for risk, effective environmental screening and active governance [6]. The most important characteristics of organizational agility as identified by PMI [26] were:

Flexibility and adaptability, 2. Open communication, 3. Openness to change, 4. Empowered team members, 5. Experiential learning, 6. Rapid decision making, 7. A strong customer focus

The results of this study to support the objective 1 of this study found the leading top eight characteristics of organizational agility were:

1. Open communications, 2. Flexible and adaptable, 3. Transparency in decision making, 4. Empowered team members, 5. Openness to change, 6. Committed to Agility, 7. Continuous learning from experience, 8. Self-aware and honest

### 3.2 Practices of organisational agility

Improving organizational agility requires a number of ongoing practices including continuous communication, collaboration, engagement and providing support within organisations.

A PMI's report (p. 6) [27] identifies the six foundational practices of organizational agility including

1. Responding quickly to strategic opportunities, 2. Shortening production/review/decision cycles, 3. Eliminating organizational silos, 4. Aligning new business capabilities to strategy, 5. Integrating the voice of the customer and 6. Focusing on change management.

In this study respondents were asked about the following 27 practices that have been found by PMI [28] that support organizational agility:

1. Focus on change management, 2. Application of iterative project management concepts to portfolio management, 3. Use of program management practices, 4. Focus on resource management, 5. Focus on risk management, 6. Focus on lean practices and value, 7. Project task simplification, 8. Quick response to strategic opportunities, 9. Shorter production/review/ decision cycles, 10. Elimination of organisational silos, 11. Use of Project Portfolio Management practices, 12. Integrates voice of the customer, 13. Use of iterative or incremental project management practices, 14. Interdisciplinary project teams, 15. Contingency planning, 16. Leverages technology, 17, Empirical (real-time) project management, 18. Matrix management, 19. Use of models, pilots and simulation, 20. Question assumptions, 21. Assessment of disruptive technological or other changes, 22. Increased environmental scanning, 23. Growth by acquisition, 24. Focus on innovation, 25. Standardisation of project management practices, 26. Progressively elaborated and active use of Project Business Cases, 27. Use of Business Cases for post implementation reviews and benefits management

The results of this study to support objective 1 of this study found the leading top eight practices needed for organizational agility were:

1. Quick response to strategic opportunities, 2. Elimination of organisational silos, 2. Focus on change management , 3. Shorter production/review/decision cycles, 4. Integrate voice of the customer, 5. Interdisciplinary project teams, 5. Leverages technology, 6. Assessment of disruptive technological or other changes

### 3.3 Predicting agility through artificial intelligence

The data collection for this research included a comprehensive inclusivity from 44 different industry practitioners from a range of industrial sectors in Australia (including private sector

organizations, public sector, services organizations, Universities and not for profit organizations among others). Nevertheless, considering human bias and limitation in providing real-time data responses, however, it was still not enough to train Artificial Intelligence (AI) or Machine Learning (ML) models. Having high-quality, precise though comprehensive, and large dataset is important to train a machine learning algorithm effectively, as without it even the most effective ML model may fail. Hence, it is vital to have a comprehensive dataset to provide accurate and precise predictions.

Organizational success depends on its capacity to mould according to market dynamics in order to maintain a competitive edge. Organizational Agility (OA) is the capacity to swiftly adapt to changes and carry out necessary activities to keep up the growth pace. It entails a variety of traits and behaviours for example, rapid and decentralised decision making; self-aware, honest, and customer-oriented talent development; active, competitive and agile team members; catalyst leadership and continuous learning from past experience. All these traits are measurable (as detailed in Section 2) from real-time responses sought from the selected/targeted audience; however, comprehensive and consistent information is important to formulate trends and eventually predict the most probable outcome to remain vigilant and competitive in the market. Hence, this research gathers required data, analyses the necessary data parameters, uses synthetic data extension, and eventually applies machine learning models to predict the characteristics and practices that are required to adopt in order to maintain organizational agility.

A recent article in The Economist [44] has shown that data is a new and expensive asset. Companies and researchers try to gather a large volume of quality data for data analytics to provide insights and trends. Machine Learning (ML) techniques rely on extensive training and validation datasets [40]. ML maps the input data to output data, for which enough data is needed to capture the relationships between input and output features. Even with the knowledge of the importance of data, data analysts sometimes encounter limited data. The difficulty of data collection, time and privacy, and legal restrictions on data might be reasons for freely collecting data. In such cases, techniques such as data augmentation or synthetic data can assist in generating additional data. Data augmentations are generally used in images with minor transformations such as zooms, flips, cropping, shifts, blur, rotation, translation etc. In comparison, data synthesis is a technique to generate artificial data that mimics observations to train ML models when actual data is difficult or expensive to get [45].

Synthetic Data Vault (SDV) Python package [46] was applied to the existing data to generate additional synthetic data in this study. SDV is a popular synthetic data generator and validation framework using multivariate cumulative distribution functions or Generative Adversarial Networks (GANs). In machine learning (ML) paradigm, the GANs are utilised to increase accuracy of data points, as they are based on two concurrent neural networks, which compete with one another to generate more accurate predictions. The data was first organised according to the scenarios (scenario-1, scenario-2). Each scenario comprises of three categories of outputs with respect to the degree of organisational agility (1, 2, or 3). Two hundred synthetic data points were created by initiating the Gaussian Copula model and fitting it with the data. To test the goodness of fit of the synthetic data, evaluate (metrics = [KSTest]) method with metrics Inverted Kolmogorov-Smirnov D statistic (KSTest) from SDV class was used to compare the real dataset with the sample dataset.

The KSTest score is normalised between 0 and 1 with the target to maximise the score. From the result above, the generated samples are good and suitable for the purpose of this study.

**Scenario-1 Results:** Hadjinicolaou et al [6] explored diverse characteristics of organizational agility and how diverse combinations existed within organizations at different levels of

strategic management maturity. The dataset utilised in this study, as detailed in Sections 1 and 2, is based on 21 factors related to organizational agility and project portfolio management, which have been categorised into three distinct data groups (i.e., Group 1, Group 2, and Group 3) by utilising Canonical analysis after a thorough synthetic data review. The findings, [6] reveal a considerable positive link between organizational agility and SPM maturity within the groups studied.

In order to provide a comprehensive analysis and to increase the readability, the experimental process has been categorised into two distinct scenarios as detailed above in section 3.3 and both of these scenarios have been processed through effective Machine Learning (ML) techniques: Support Vector Machine (SVM), Decision Tree (DT), k-Nearest Neighbours (KNN), Gradient Boosting Machine (GBM), Random Forest (RF), Naïve Bayes (NB) and Logistic Regression (LR). In Scenario 1, only the most relatable 92 responses are utilised including organizational type, organizational size, industry sectors, organizational support structures, level of internal and external change, organizational characteristics, traits, and practices of Organizational Agility (OA). The experimental results reflect the efficiency of the aforementioned ML techniques SVM, DT, KNN, GBM, RF, NB and LR as shown in Table 1, which also presents the group-wise accuracy percentages in each respective group (e.g., Group 1, Group 2, and Group 3).

The overall high test accuracy percentages confirm the effectiveness of the ML algorithms utilised in this study, which reflects that RF provides the best possible overall outcome with the accuracy of 97.674%, that is slightly better than SVM, GBM and KNN. Nonetheless, the results vary in different groups, however, there is no test accuracy that is less than 81.818%.

To demonstrate the performance of the ML models in more details, the related test confusion matrices are depicted in Table 2. A confusion matrix provides a comparison of the actual target values with the predicted values to illustrate the performance of machine learning models. The confusion matrices used for the ML models employed in this study as highlighted in Table 2, show the numbers of the correct and incorrect decisions made in each group.

Furthermore, test models' sensitivity, specificity and F1 scores for the three groups have been reported in Table 3 below.

The results show that RF has had the highest overall sensitivity, specificity and F1 score and KNN, GBM and SVM have had scores which are relatively comparable with RF.

The results of these test models yield a substantial output as depicted in Fig 4. A closer look at Fig 4 explains that in most of the cases the model has predicted the values quite accurately, where red circles show the model's test output and blue dots represent the desired test output. As it is depicted in Fig 4, there are just a few cases where the test outcome is different from the desired values (such as (9,2), (3,2), (17,2) and (18,3)).

**Table 1. Accuracy percentages of the test models.**

| Sr# | Method | Accuracy Percentages | Group-wise Accuracy Percentages | | |
|---|---|---|---|---|---|
| | | | Group 1* | Group 2* | Group 3* |
| 1 | SVM | 95.348 | 90.697 | 97.619 | 97.727 |
| 2 | DT | 91.472 | 93.023 | 95.238 | 86.363 |
| 3 | KNN | 96.899 | 95.348 | 97.619 | 97.727 |
| 4 | GBM | 94.573 | 100 | 92.857 | 90.909 |
| 5 | RF | 97.674 | 100 | 97.628 | 97.445 |
| 6 | NB | 93.798 | 100 | 95.238 | 86.363 |
| 7 | LR | 89.147 | 93.012 | 92.867 | 81.818 |

*Group1, 2 or 3 categorise the degree of organisational agility (1, 2, or 3).

**Table 2. Test confusion matrices.**

| Sr# | Method | Test Confusion Matrices |
|---|---|---|
| 1 | Support Vector Machine | $\begin{bmatrix} 39 & 3 & 1 \\ 1 & 41 & 0 \\ 0 & 1 & 43 \end{bmatrix}$ |
| 2 | Decision Tree | $\begin{bmatrix} 40 & 1 & 2 \\ 1 & 40 & 1 \\ 3 & 3 & 38 \end{bmatrix}$ |
| 3 | k-Nearest Neighbours | $\begin{bmatrix} 41 & 1 & 1 \\ 1 & 41 & 0 \\ 0 & 1 & 43 \end{bmatrix}$ |
| 4 | GBM | $\begin{bmatrix} 43 & 0 & 0 \\ 2 & 39 & 1 \\ 1 & 3 & 40 \end{bmatrix}$ |
| 5 | RF | $\begin{bmatrix} 43 & 0 & 0 \\ 1 & 41 & 0 \\ 0 & 2 & 42 \end{bmatrix}$ |
| 6 | NB | $\begin{bmatrix} 43 & 0 & 0 \\ 1 & 40 & 1 \\ 3 & 3 & 38 \end{bmatrix}$ |
| 7 | LR | $\begin{bmatrix} 40 & 1 & 2 \\ 1 & 39 & 2 \\ 3 & 5 & 36 \end{bmatrix}$ |

The overall accuracy percentage for SVM (as presented in Table 3) is also depicted in Fig 5A, which reflects the performance of the test models to suffice the experiments.

The accuracy percentages in each group are demonstrated in Fig 5B. Colour distribution distinguishes between the test mistakes and the correct predictions for the respective groups e.g., the bold red segment reflects the test mistakes in Group1, and the light green segment shows the correct decisions in the same group. Similarly, the small purple segment in the second inner circle reflects the test mistakes, and the bigger dark green segment shows the correct predictions for Group2 and the same goes for Group3.

*Decision Tree (DT)*: For the test scenario as detailed above, Decision Tree (DT) method has also been applied and results have been generated as per the following Test Confusion Matrix highlighted in Table 4. Fig 6 represents the results of the overall test outcomes using DT, which reflects that in most of the cases the model has predicted the values quite accurately. Red circles in Fig 6 show the model's test outputs and blue dots represent the desired test outputs. It's just a few cases the developed model has made a mistake in identifying the correct group.

The overall accuracy percentage for Decision Tree (DT) (as presented in Table 1) is depicted in Fig 7A which reflects the performance of the test models.

The accuracy percentages in each group are demonstrated in Fig 7B. Colour distribution distinguishes between the test mistakes and the correct predictions for the respective groups e.g., the bold red segment reflects the test mistakes in Group1, and the light green segment shows the correct decisions in the same group. Similarly, the small purple segment in the

**Table 3. Sensitivity, specificity and F1 scores of the test models for Scenario1.**

| Sr# | Method | Sensitivity | | | Specificity | | | F1 | | |
|---|---|---|---|---|---|---|---|---|---|---|
| | | Group1* | Group 2* | Group 3* | Group1* | Group 2* | Group 3* | Group1* | Group 2* | Group 3* |
| 1 | SVM | 97.50 | 91.11 | 97.72 | 98.88 | 95.40 | 98.82 | 93.97 | 94.24 | 97.72 |
| 2 | DT | 90.90 | 90.90 | 92.68 | 95.34 | 95.40 | 96.47 | 91.94 | 93.01 | 89.62 |
| 3 | KNN | 97.76 | 95.34 | 97.72 | 98.83 | 97.70 | 98.82 | 96.53 | 96.46 | 97.72 |
| 4 | GBM | 93.47 | 92.85 | 97.56 | 96.51 | 96.55 | 98.82 | 96.62 | 92.85 | 94.11 |
| 5 | RF | 97.72 | 95.34 | 100 | 98.83 | 97.70 | 100 | 98.84 | 96.46 | 97.67 |
| 6 | NB | 91.48 | 93.02 | 97.43 | 95.34 | 96.55 | 98.82 | 95.55 | 94.11 | 91.61 |
| 7 | LR | 90.90 | 86.86 | 90.00 | 95.34 | 93.10 | 95.29 | 91.94 | 89.75 | 85.59 |

*Group1, 2 or 3 categorise the degree of organisational agility (1, 2, or 3).

second inner circle reflects the test mistakes, and the bigger dark green segment shows the correct predictions for Group2 and the same goes for Group3.

*k-Nearest Neighbours (KNN)*: For the test scenario as detailed above, k-Nearest Neighbours (KNN) along with the Euclidean distance have been applied. The Test Confusion Matrix is given in Table 1. Fig 8 represents the results of the overall test outcome applying KNN, which reflects that in most of the cases the model has predicted the values quite accurately. As mentioned above, red circles show the model's test outputs and blue dots represent the desired test outputs. It is just a few cases the model has made mistakes, e.g., points (7,1) and (18,2).

Furthermore, the overall accuracy percentage for KNN (as presented in Table 1) is depicted in Fig 9A that reflects the good performance of the test model.

The accuracy percentages in each group are demonstrated in Fig 9B. Colour distribution distinguishes between the test mistakes and the correct predictions for the respective groups e.g., the bold red segment reflects the test mistakes in Group1, and the light green segment shows the correct decisions in the same group. Similarly, the small purple segment in the second inner circle reflects the test mistakes, and the bigger dark green segment shows the correct predictions for Group2 and the same goes for Group3.

The GBM test outcomes for Group1, Group2 and Group3 have been demonstrated in Fig 10 below.

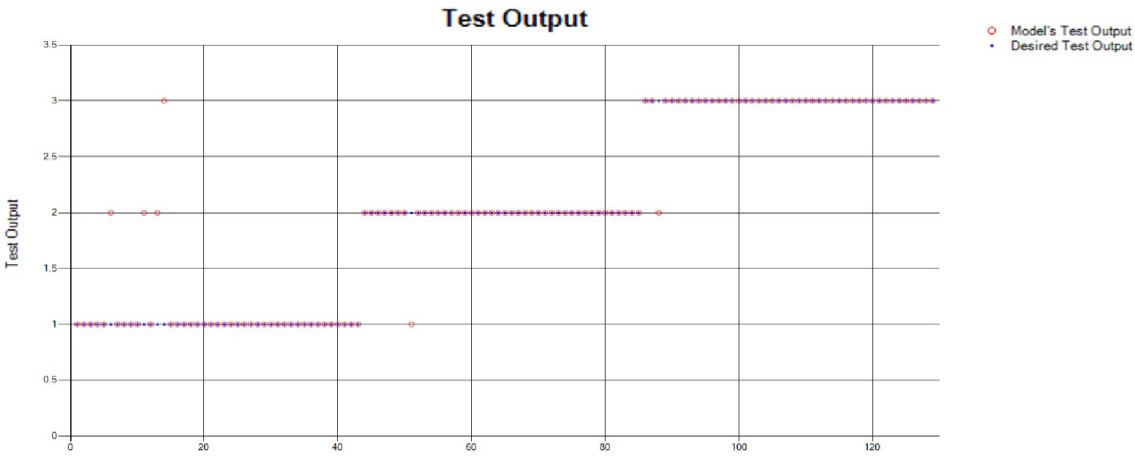

**Fig 4. Scenario 1 SVM test outcome.**

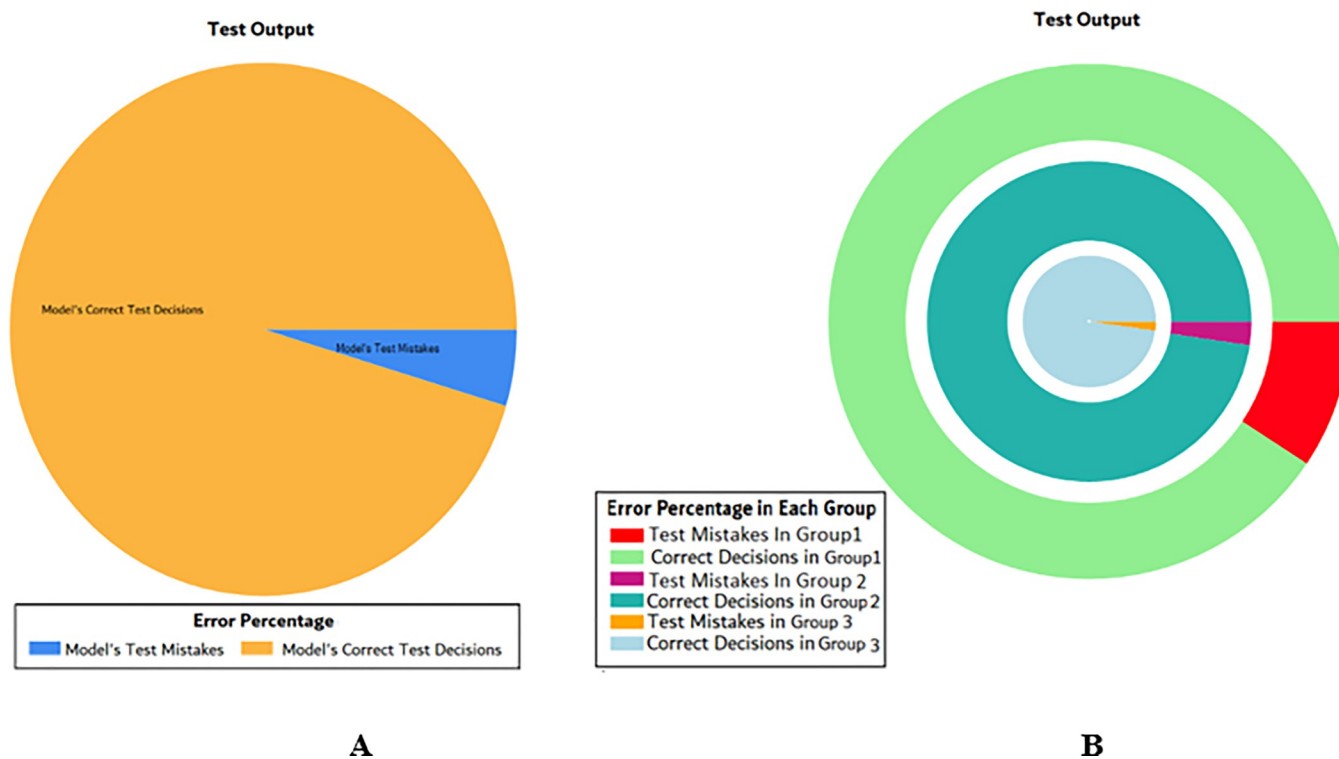

**Fig 5. Scenario 1 SVM test overall accuracy percentage and distribution each group.**

As Fig 10 shows, there have been only 7 mistakes and the rest of the outcomes have been categorised correctly.

Fig 11A shows the GBM's overall test accuracy percentage whereas the test accuracy percentages within each group for GBM have been demonstrated in Fig 11B.

The test outcomes for RF are shown in Fig 12. There have been only 3 test mistakes and the rest of the decisions have for the test dataset have been made correctly.

The overall accuracy percentage for RF and the accuracies for each groups are demonstrated in Fig 13A and 13B respectively. RF has had a very good performance compared to the other methods applied in the dataset for Scenario1.

Fig 14 outlines the test outcomes produced by NB. There have been 8 mistakes and the rest of the decisions have been made correctly.

The test accuracy for each group and the overall test accuracy percentage for NB have been demonstrated in Fig 15. It is understandable that NB hasn't been able to provide the results as good as RF or GBM.

**Table 4. Scenario-wise KSTest* scores.**

| Scenario 1 | | Scenario 2 | |
|---|---|---|---|
| Dataset | KSTest score | Dataset | KSTest score |
| Group 1** | 0.685 | Group 1** | 0.683 |
| Group 2** | 0.856 | Group 2** | 0.875 |
| Group 3** | 0.890 | Group 3** | 0.900 |

*Kolmogorov-Smirnov D statistic Test.

**Group1, 2 or 3 categorise the degree of organisational agility (1, 2, or 3).

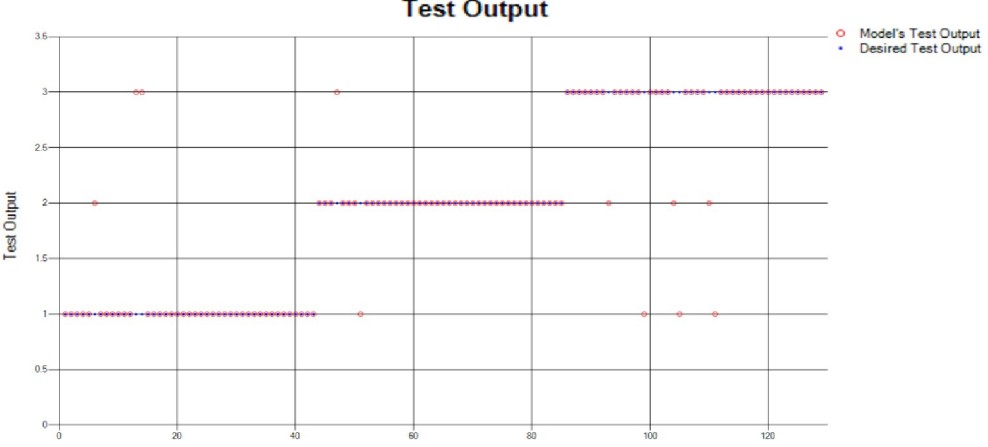

**Fig 6. Scenario 1 decision tree test outcome.**

LR's test performance is shown in Fig 16. LR has made more test mistakes in compare to the other methods applied on the Scenario1 dataset.

Fig 17A and 17B show the overall accuracy and the accuracy percentage in each group related to LR for Scenario1.

***Scenario-2 Results:*** In Scenario 2, all the 142 data responses are utilised including all the identified characteristics, traits and practices of Organizational Agility (OA). SVM, DT, KNN, GBM, RF, NB and LR have been applied to Scenario-2 as well and the simulation results reflect the efficiency of the aforementioned ML techniques as shown in Table 5. In addition, Table 5

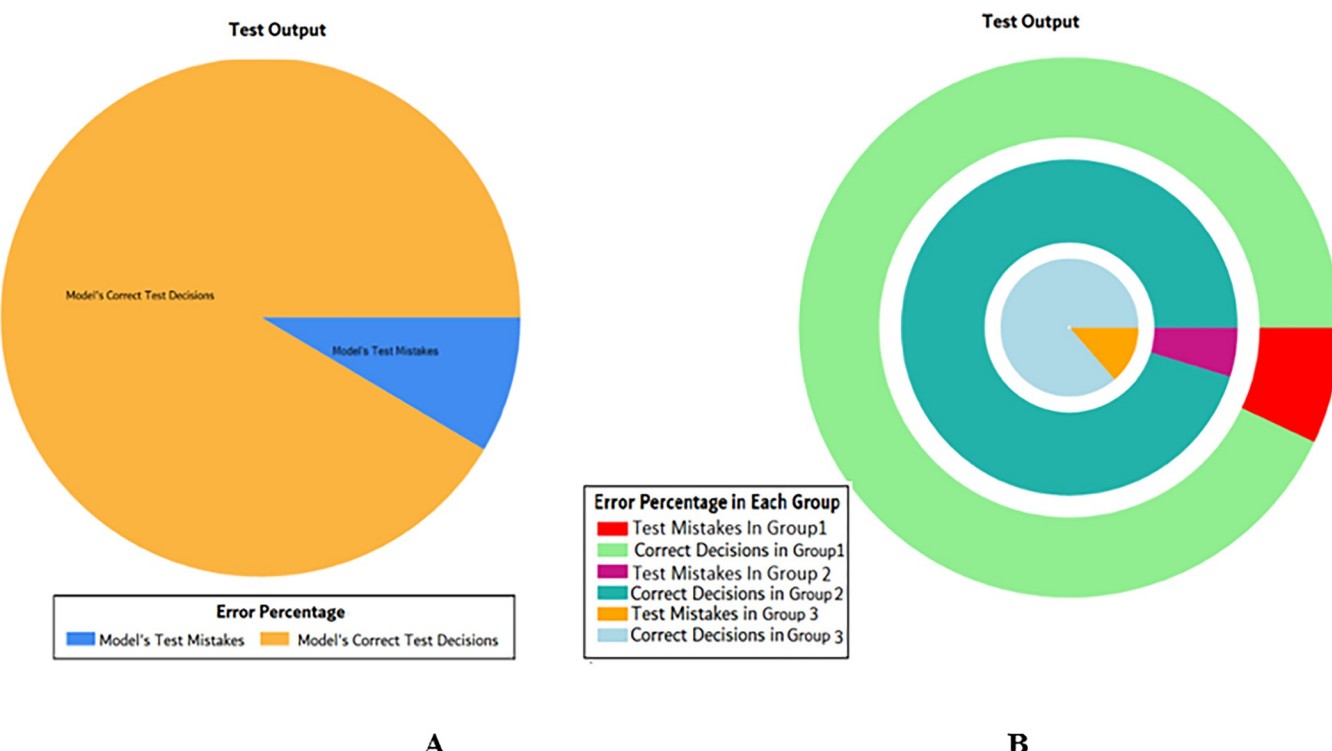

A                                                                                                    B

**Fig 7. Scenario 1 DT test overall accuracy percentage and distribution each group.**

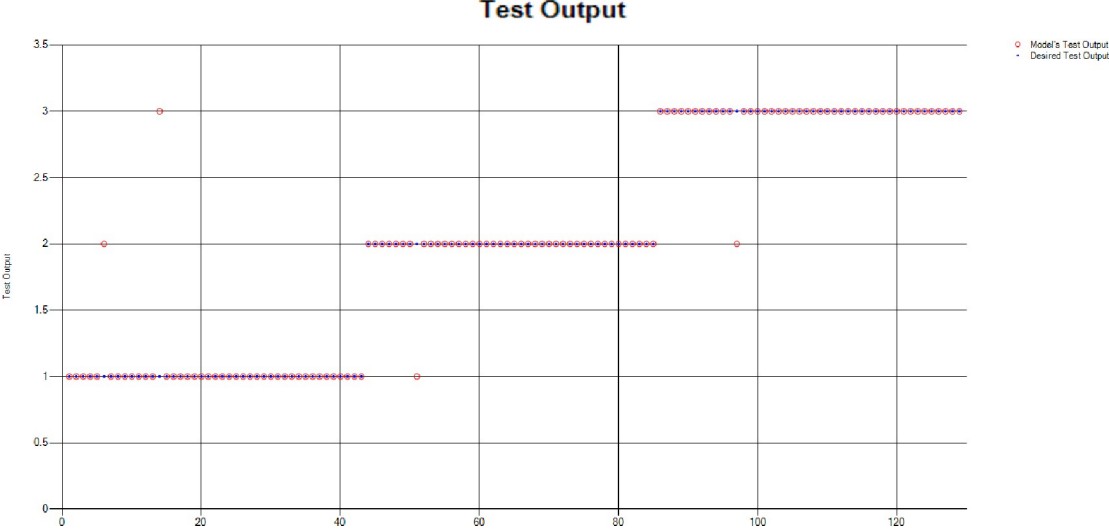

**Fig 8. Scenario 1 KNN test outcome.**

also presents the accuracy percentages in each respective group (e.g., Group 1, Group 2, and Group 3).

The overall 91+ test accuracy percentages confirm the effectiveness of the ML algorithms utilised in this study. KNN provides the best possible overall outcome with the accuracy of 99.224%, that is better than SVM, RF and other methods in this scenario.

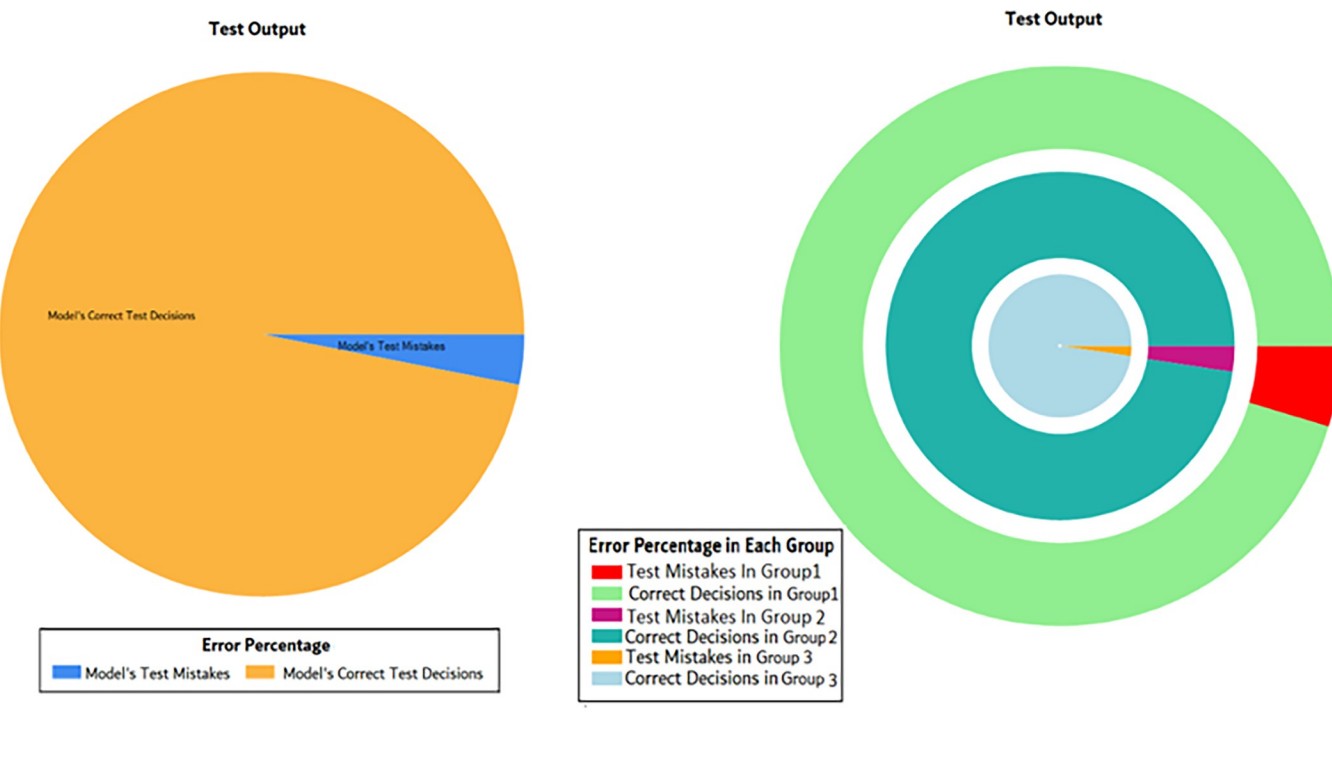

A                                                                          B

**Fig 9. Scenario 1 KNN test overall accuracy percentage and distribution each group.**

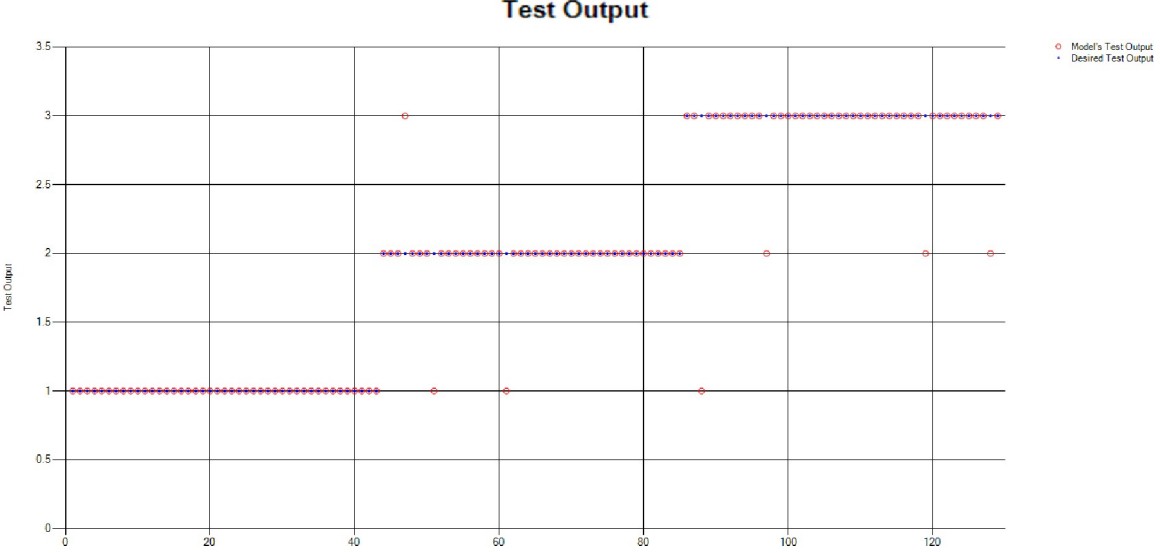

**Fig 10. Scenario 1 GBM test outcome.**

Confusion matrices are used to evaluate the performance of the ML models since they provide a clear comparison between the actual target values with the predicted values. The applied ML models' confusion matrices are shown in Table 6:

Furthermore, test models' sensitivity, specificity and F1 scores for the three groups have been reported in Table 7 below.

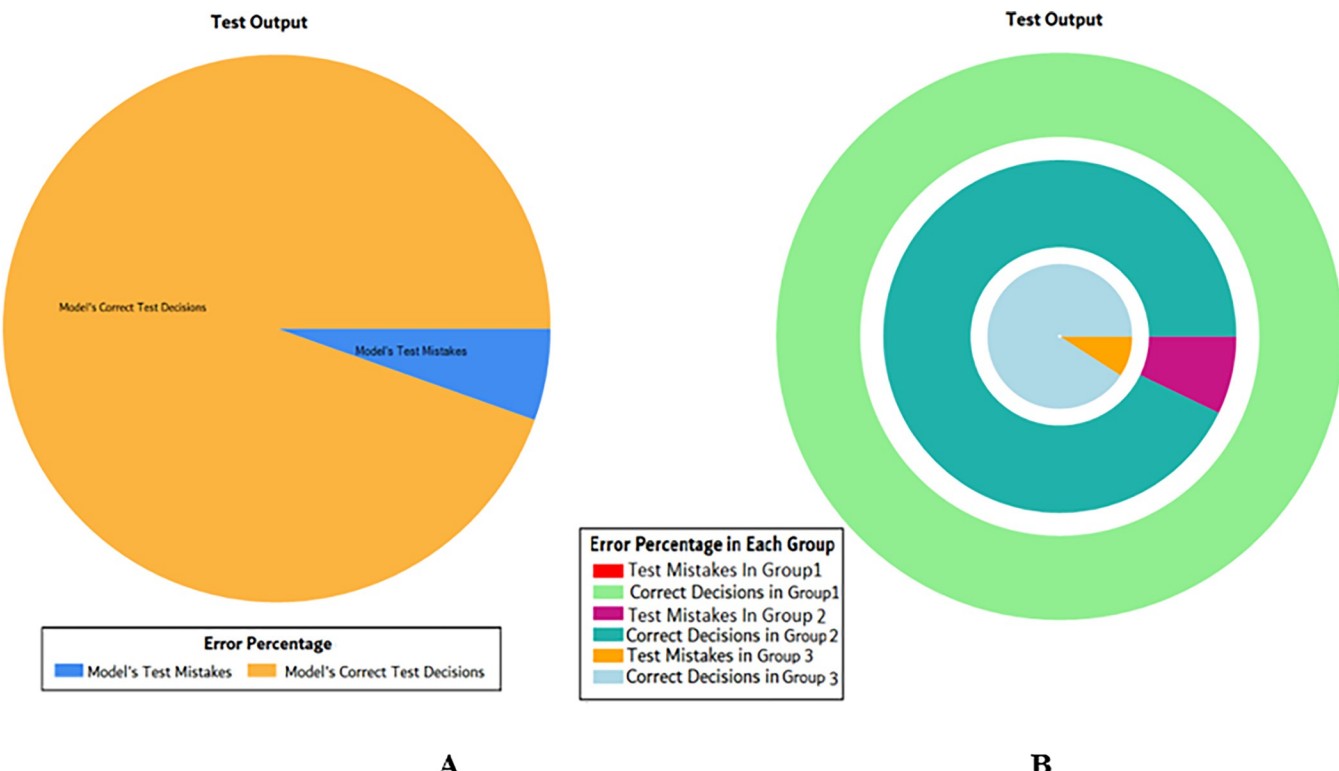

A                                                                    B

**Fig 11. Scenario 1 GBM test overall accuracy percentage and distribution each group.**

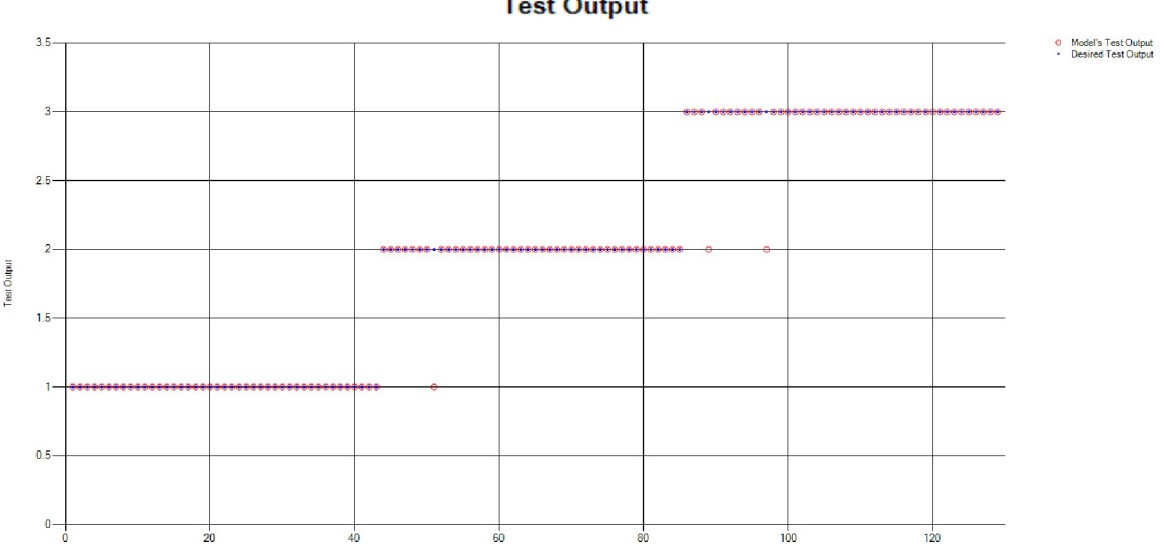

**Fig 12. Scenario 1 RF test outcome.**

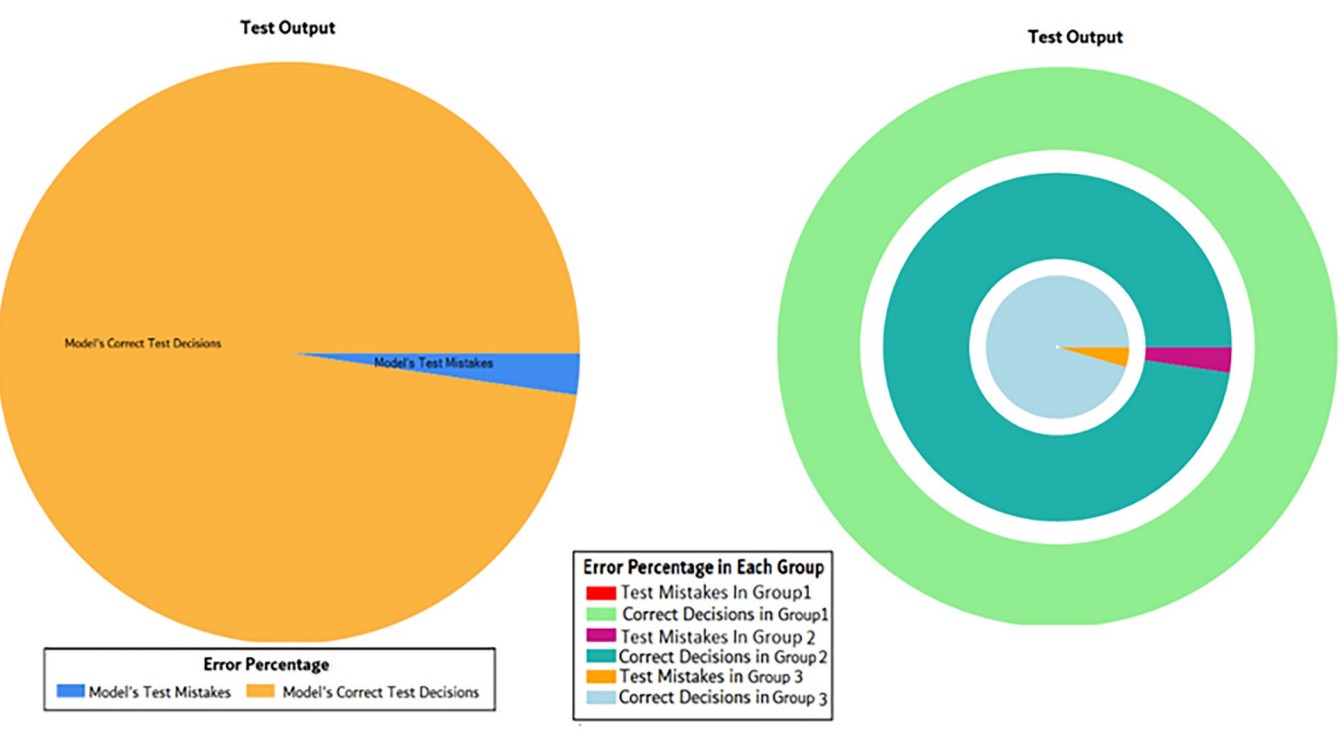

**A**                                                     **B**

**Fig 13. Scenario 1 RF test overall accuracy percentage and distribution each group.**

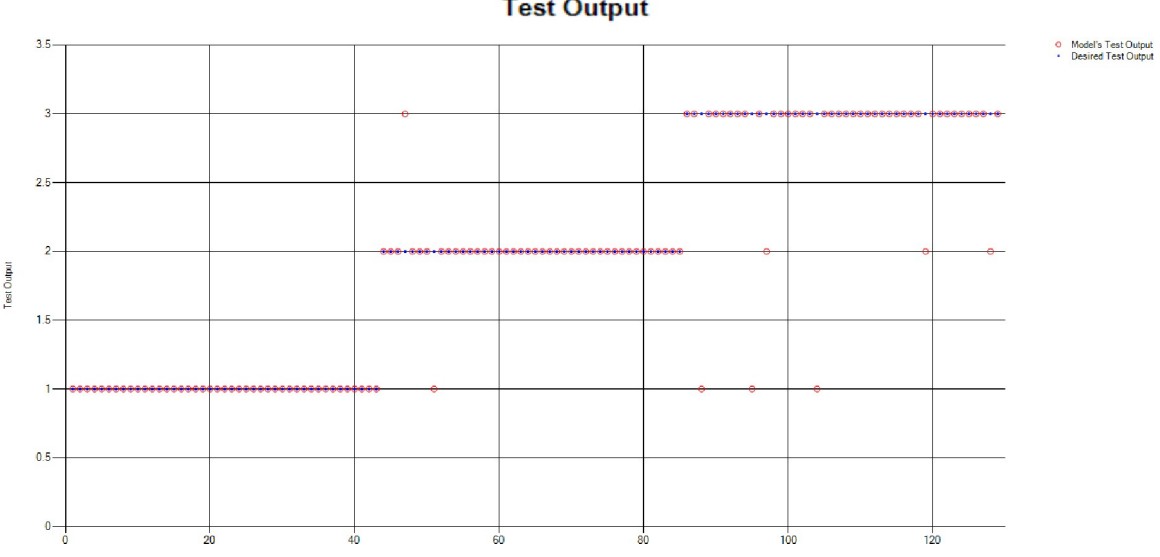

**Fig 14. Scenario 1 NB test outcome.**

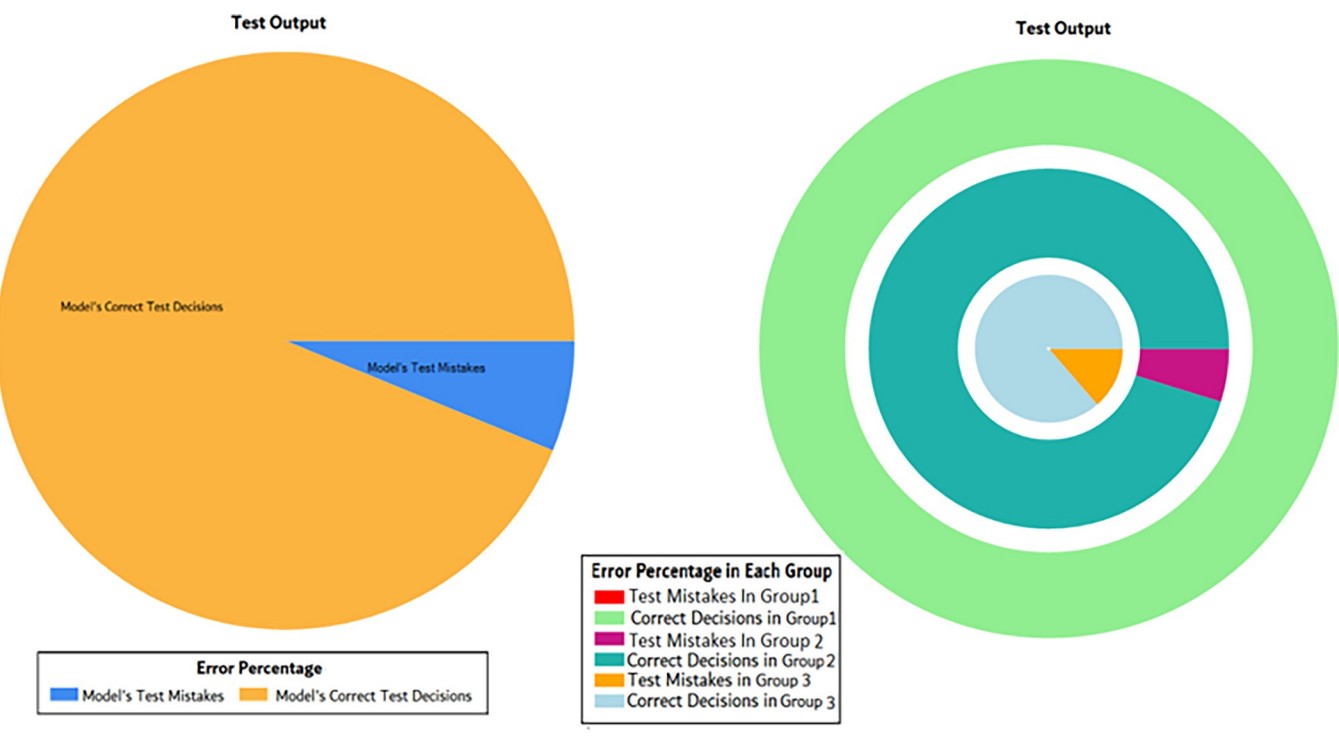

**Fig 15. Scenario 1 NB test overall accuracy percentage and distribution each group.**

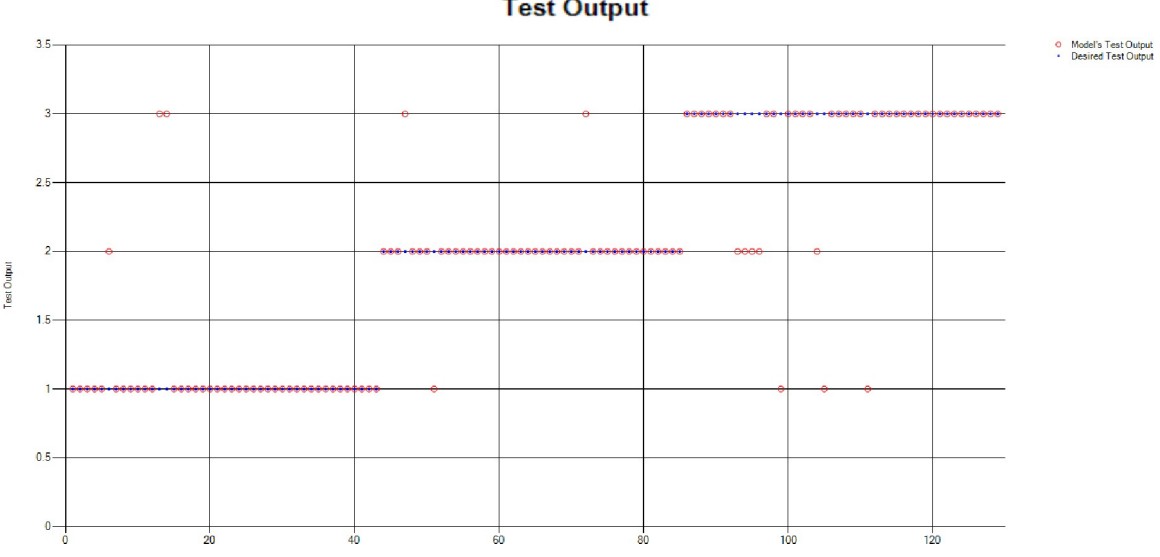

**Fig 16. Scenario 1 LR test outcome.**

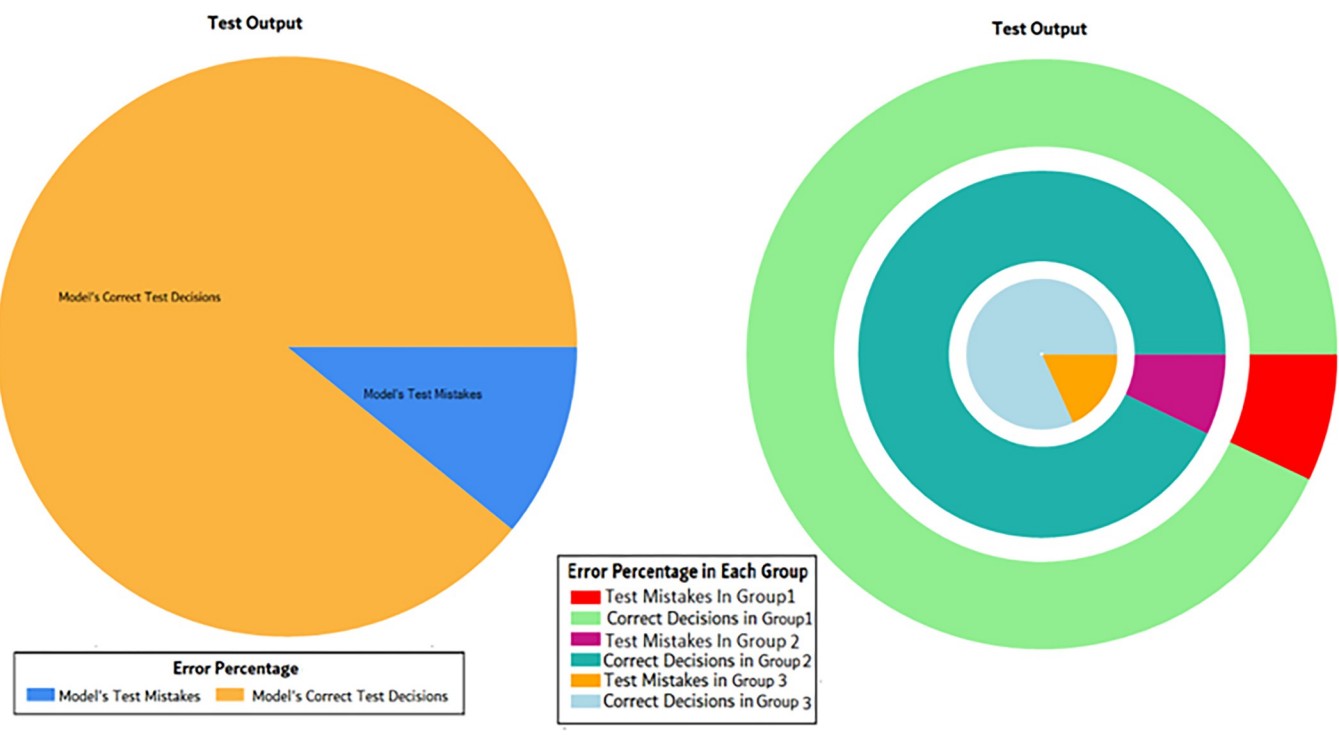

**Fig 17. Scenario 1 LR test overall accuracy percentage and distribution each group.**

**Table 5. Accuracy percentages of the test models.**

| Sr# | Method | Accuracy Percentages | Group-wise Accuracy Percentages | | |
|-----|--------|---------------------|------------|------------|------------|
| | | | Group1* | Group2* | Group 3* |
| 1 | SVM | 96.899 | 93.023 | 97.619 | 100 |
| 2 | DT | 91.472 | 90.697 | 92.857 | 90.909 |
| 3 | KNN | 99.224 | 100 | 97.619 | 100 |
| 4 | GBM | 96.899 | 95.348 | 97.624 | 97.717 |
| 5 | RF | 98.449 | 97.674 | 97.619 | 100 |
| 6 | NB | 90.697 | 90.697 | 90.476 | 90.908 |
| 7 | LR | 86.046 | 90.706 | 88.093 | 79.545 |

*Group1, 2 or 3 categorise the degree of organisational agility (1, 2, or 3).

The results of these test models are depicted in Fig 18. The results in Fig 18 explain that in most of the cases the model has predicted the values quite accurately, where red circles show the model's test output and blue dots represent the desired test output.

The overall accuracy percentage for SVM (as presented in Table 5) is shown in Fig 19A which reflects the performance of the test models.

The accuracy percentages in each group are demonstrated in Fig 19B. Colour distribution distinguishes between the test mistakes and the correct predictions for the respective groups e.g., the bold red segment reflects the test mistakes in Group1, and the light green segment

**Table 6. Test confusion matrices.**

| Sr# | Method | Test Confusion Matrix |
|-----|--------|----------------------|
| 1 | Support Vector Machine | $\begin{bmatrix} 40 & 2 & 1 \\ 1 & 41 & 0 \\ 0 & 0 & 44 \end{bmatrix}$ |
| 2 | Decision Tree | $\begin{bmatrix} 39 & 3 & 1 \\ 1 & 39 & 2 \\ 0 & 4 & 40 \end{bmatrix}$ |
| 3 | k-Nearest Neighbours | $\begin{bmatrix} 43 & 0 & 0 \\ 1 & 41 & 0 \\ 0 & 0 & 44 \end{bmatrix}$ |
| 4 | GBM | $\begin{bmatrix} 41 & 1 & 1 \\ 1 & 41 & 0 \\ 0 & 1 & 43 \end{bmatrix}$ |
| 5 | RF | $\begin{bmatrix} 42 & 0 & 1 \\ 1 & 41 & 0 \\ 0 & 0 & 44 \end{bmatrix}$ |
| 6 | NB | $\begin{bmatrix} 39 & 3 & 1 \\ 1 & 38 & 3 \\ 0 & 4 & 40 \end{bmatrix}$ |
| 7 | LR | $\begin{bmatrix} 39 & 4 & 0 \\ 1 & 37 & 4 \\ 0 & 9 & 35 \end{bmatrix}$ |

**Table 7. Sensitivity, specificity and F1 scores of the test models for Scenario2.**

| Sr# | Method | Sensitivity | | | Specificity | | | F1 | | |
|---|---|---|---|---|---|---|---|---|---|---|
| | | Group1* | Group 2* | Group 3* | Group1* | Group 2* | Group 3* | Group1* | Group 2* | Group 3* |
| 1 | SVM | 97.56 | 95.34 | 97.77 | 98.83 | 97.70 | 98.82 | 95.23 | 96.46 | 98.87 |
| 2 | DT | 97.50 | 84.78 | 93.02 | 98.83 | 91.95 | 96.47 | 93.97 | 88.63 | 91.94 |
| 3 | KNN | 97.72 | 100 | 100 | 98.83 | 100 | 100 | 98.84 | 98.79 | 100 |
| 4 | GBM | 97.61 | 95.34 | 97.72 | 98.83 | 97.70 | 98.82 | 96.46 | 96.46 | 97.72 |
| 5 | RF | 97.67 | 100 | 97.77 | 98.83 | 100 | 98.82 | 97.67 | 98.79 | 98.87 |
| 6 | NB | 97.50 | 84.44 | 90.90 | 98.83 | 91.95 | 95.29 | 93.97 | 87.35 | 90.90 |
| 7 | LR | 97.50 | 74.00 | 89.74 | 98.83 | 85.05 | 95.29 | 93.97 | 80.43 | 84.33 |

*Group1, 2 or 3 categorise the degree of organisational agility (1, 2, or 3).

shows the correct decisions in the same group. Similarly, the small purple segment in the second inner circle reflects the test mistakes, and the bigger dark green segment shows the correct predictions for Group2 and the same goes for Group3.

*Decision Tree (DT)*: Additionally, for the test scenario as detailed above, Decision Tree (DT) has been applied and results have been generated as per the following Test Confusion Matrix highlighted in Table 6. Fig 20 represents the results of the overall test outcomes using DT which reflects that in most of the cases the model has predicted the values quite accurately. Red circles in Fig 20 show the model's test outputs and blue dots represent the desired test outputs. It's just a few cases the developed model has made a mistake in identifying the correct group.

The overall accuracy percentage for Decision Tree (DT) (as presented in Table 3) is depicted in Fig 13A to illustrate the performance of the test models.

The accuracy percentages in each group are demonstrated in Fig 21B. Colour distribution distinguishes between the test mistakes and the correct predictions for the respective groups e.g., the bold red segment reflects the test mistakes in Group1, and the light green segment shows the correct decisions in the same group. Similarly, the small purple segment in the

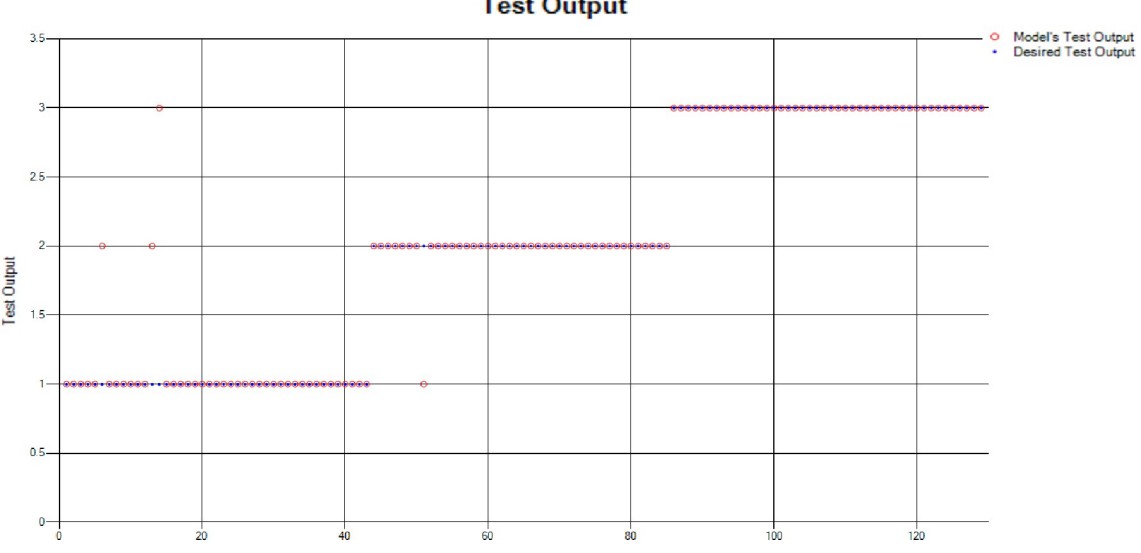

**Fig 18. Scenario 2 SVM test outcome.**

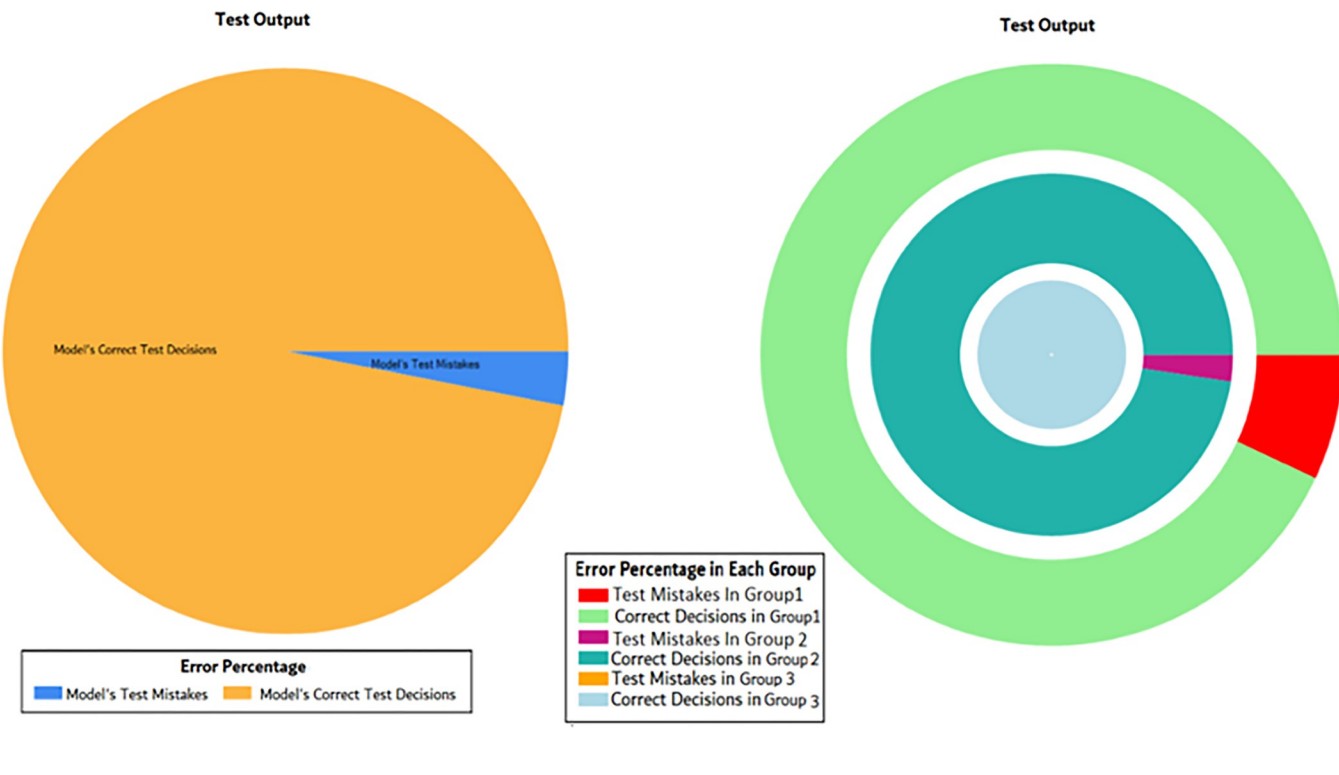

**Fig 19. Scenario 2 SVM overall accuracy percentage and distribution each group.**

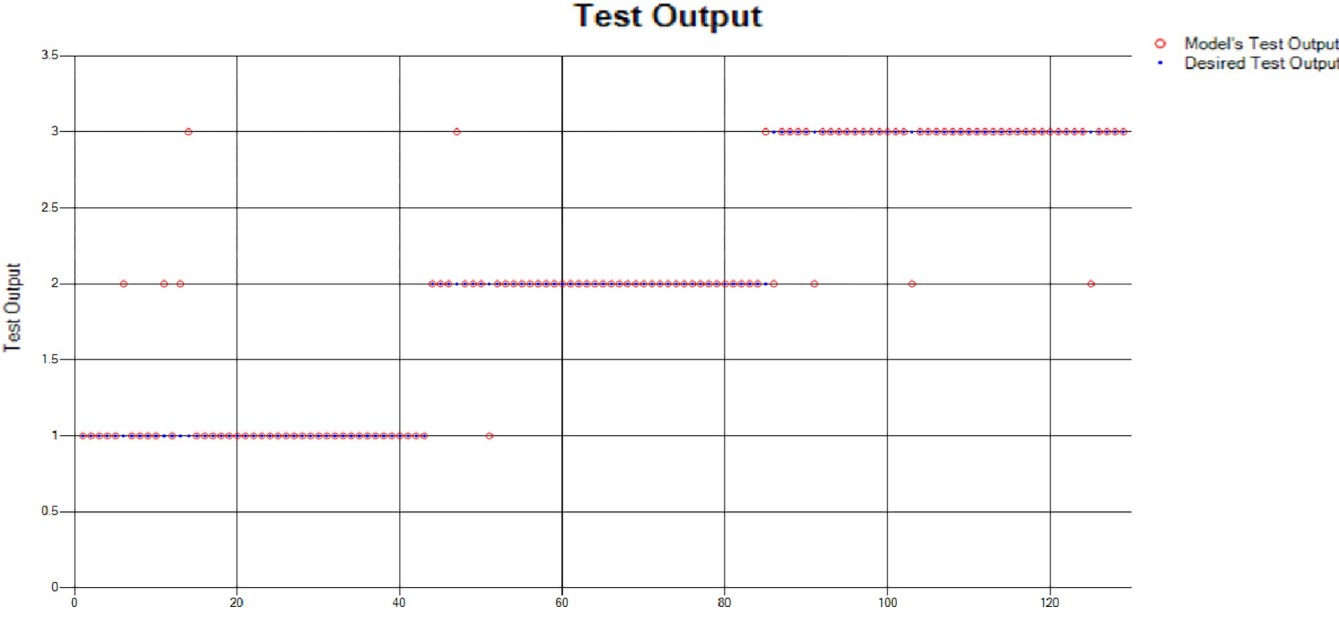

**Fig 20. Scenario 2 decision tree test outcome.**

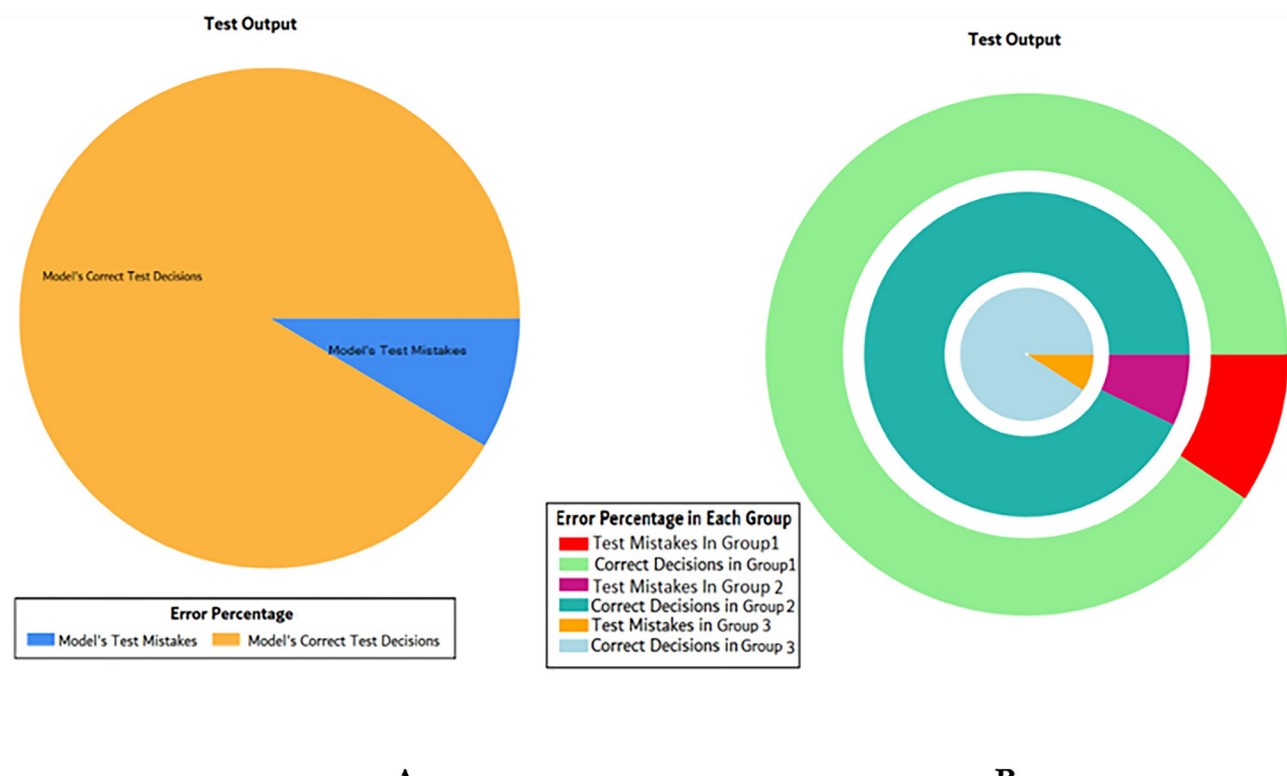

**A**

**B**

**Fig 21. Scenario 2 DT test overall accuracy percentage and distribution each group.**

second inner circle reflects the test mistakes, and the bigger dark green segment shows the correct predictions for Group2 and the same goes for Group3.

*k-Nearest Neighbours (KNN)*: For the test scenario as detailed above, k-Nearest Neighbours (KNN) along with the Euclidean distance have been applied. The Test Confusion Matrix is given in Table 6. Fig 22 represents the results of the overall test outcome after applying KNN, which reflects that in most of the cases the model has predicted the values quite accurately. As

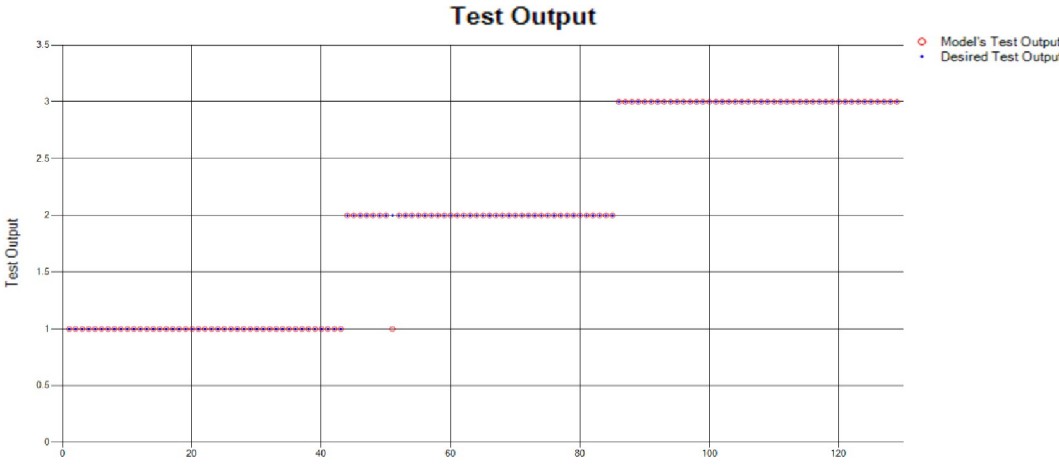

**Fig 22. Scenario 2 KNN test outcome.**

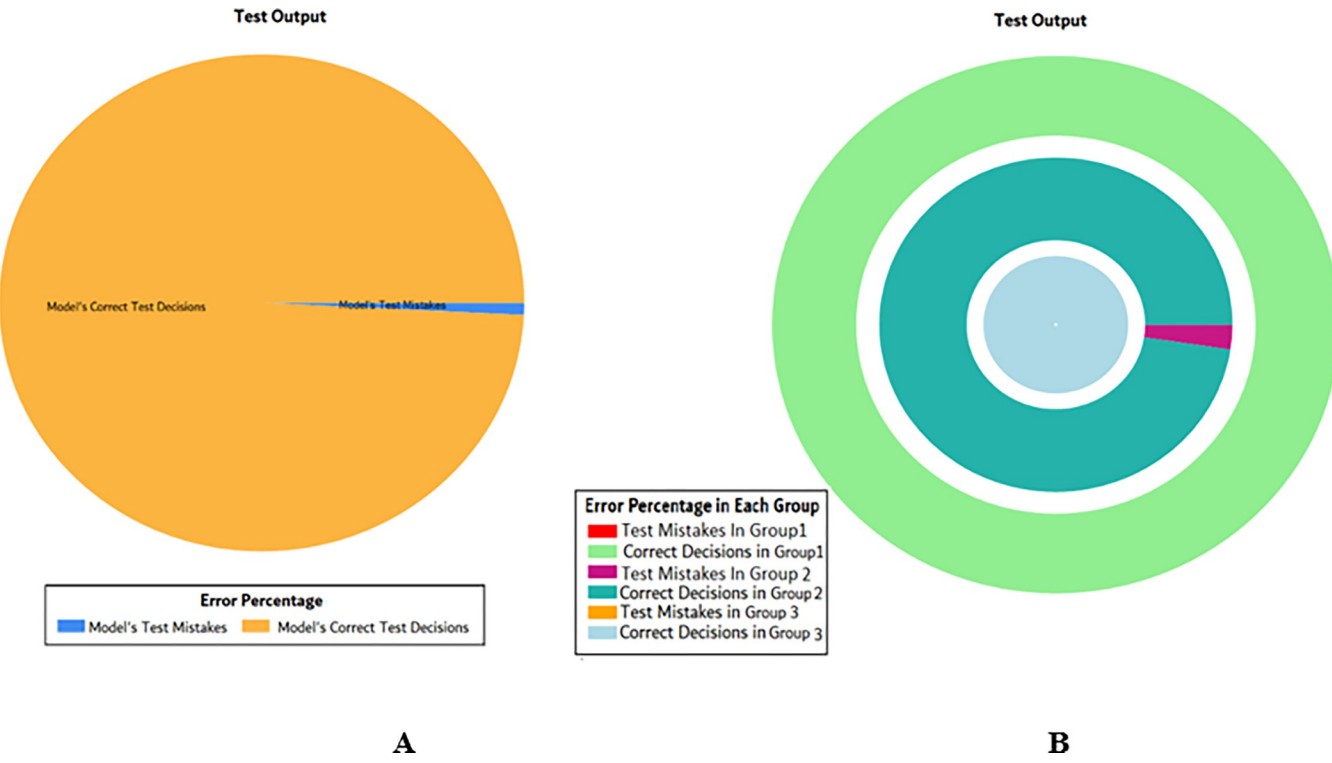

**Fig 23. Scenario 2 KNN overall accuracy percentage and distribution each group.**

mentioned above, red circles show the model's test outputs and blue dots represent the desired test outputs. It is just a few cases the model has made mistakes, e.g., points (47,1) and (47,2).

Furthermore, the overall accuracy percentage for KNN (as presented in Table 5) is depicted in Fig 23A that reflects the good performance of the test model.

The accuracy percentages in each group are demonstrated in Fig 23B. Colour distribution distinguishes between the test mistakes and the correct predictions for the respective groups e.g., the bold red segment reflects the test mistakes in Group1, and the light green segment shows the correct decisions in the same group. Similarly, the small purple segment in the second inner circle reflects the test mistakes, and the bigger dark green segment shows the correct predictions for Group2 and the same goes for Group3.

The test outcomes using GBM have been demonstrated in Fig 24. For Scenario 2, GBM has had the third best performance with respect to test accuracy after KNN, and RF (similar to SVM) among the other methods.

Similarly, the overall accuracy and the accuracies for each group are shown in Fig 25A and 25B below for GBM related to Scenario2.

The test performance for RF is demonstrated in Fig 26 for Scenario2. RF has been the second best performing method with respect to accuracy among all the other methods used. The overall test accuracy and the test accuracies for each group using RF for the second scenario are shown in Fig 27A and 27B below.

Similarly, NB's test outcome is given in Fig 28. NB hasn't performed as well as the other methods for Scenario2.

Fig 29A and 29B demonstrate the overall test accuracy and the accuracies for each group for NB.

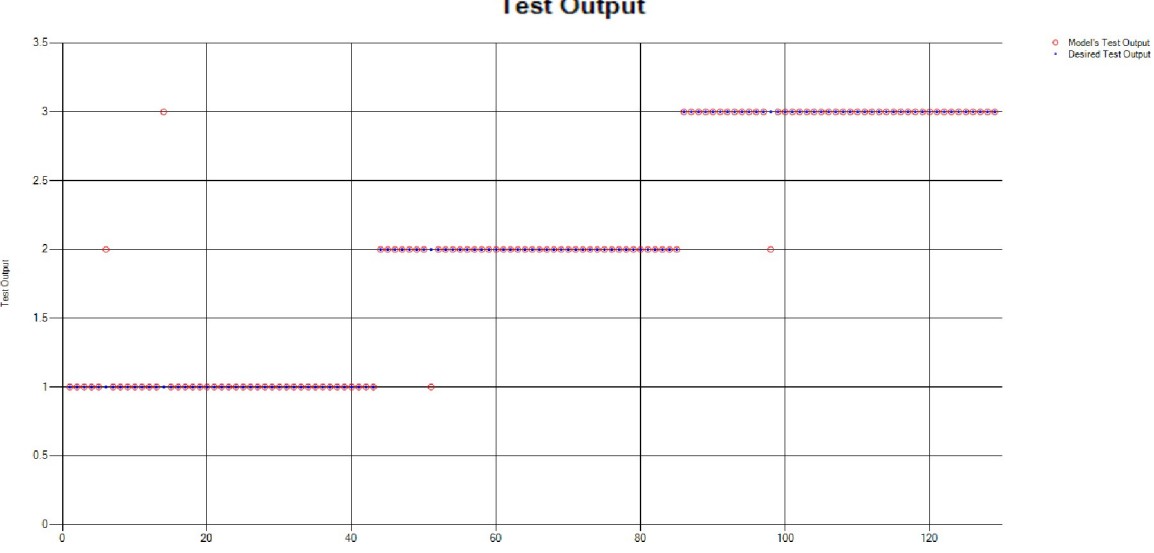

**Fig 24. Scenario 2 GBM test outcome.**

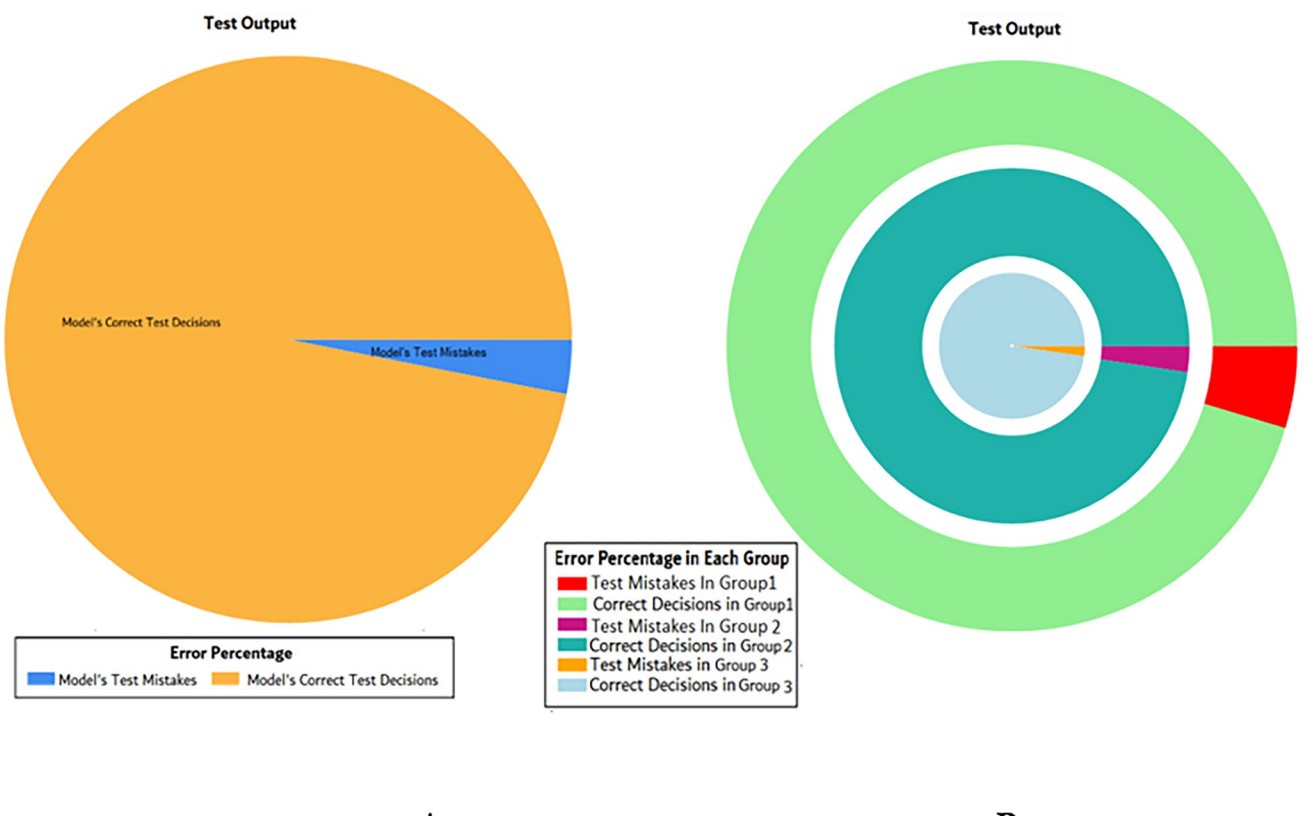

**Fig 25. Scenario 2 GBM test overall accuracy percentage and distribution each group.**

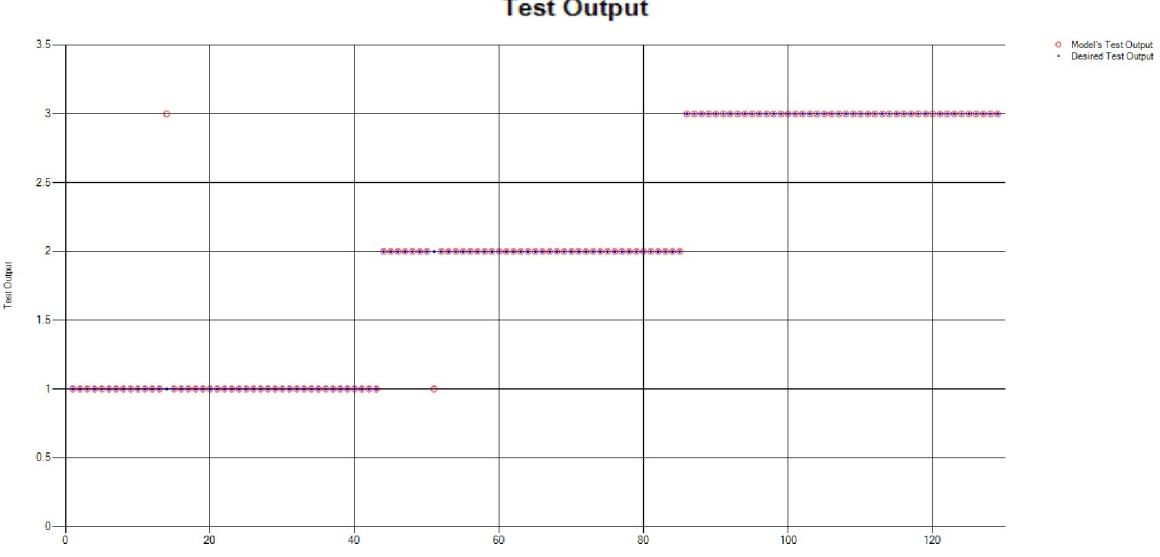

**Fig 26. Scenario 2 RF test outcome.**

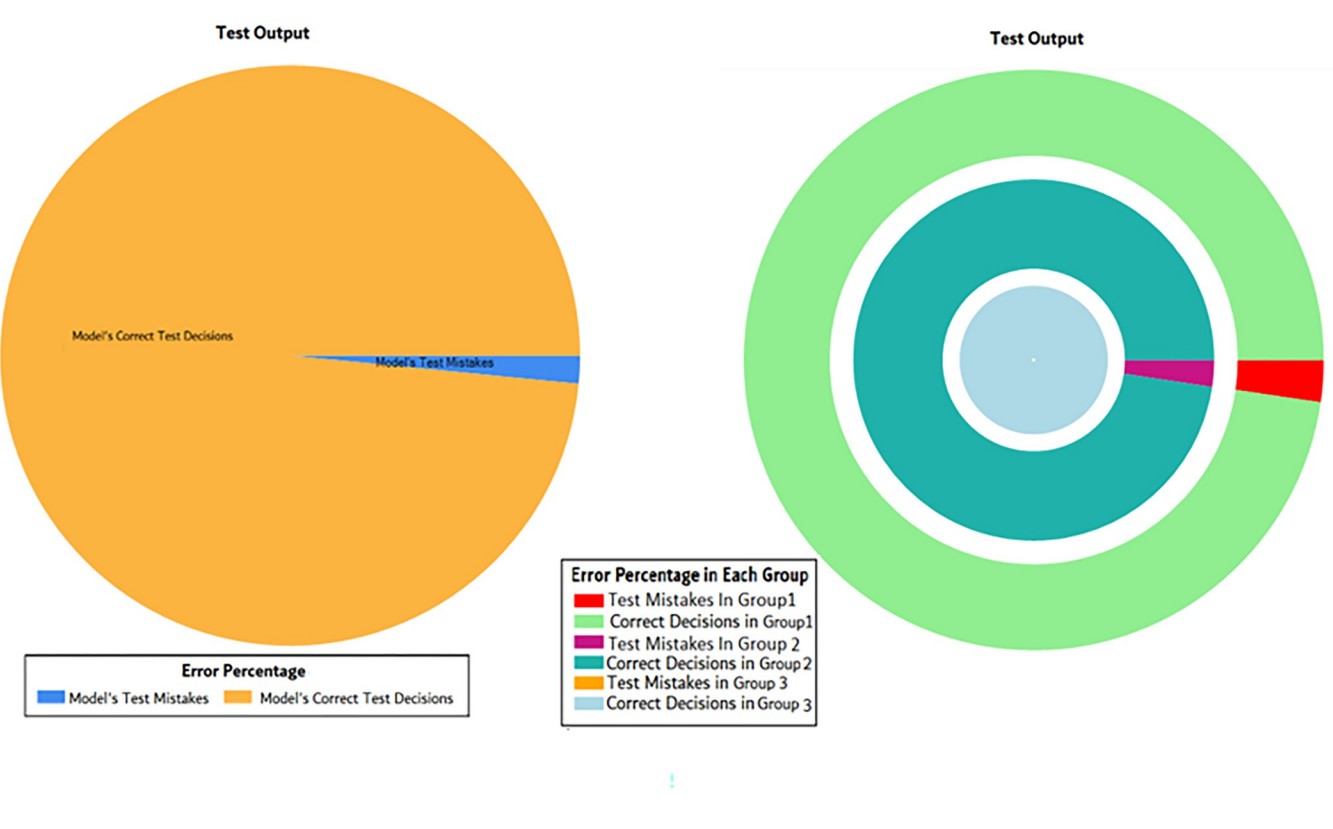

**Fig 27. Scenario 2 RF test overall accuracy percentage and distribution each group.**

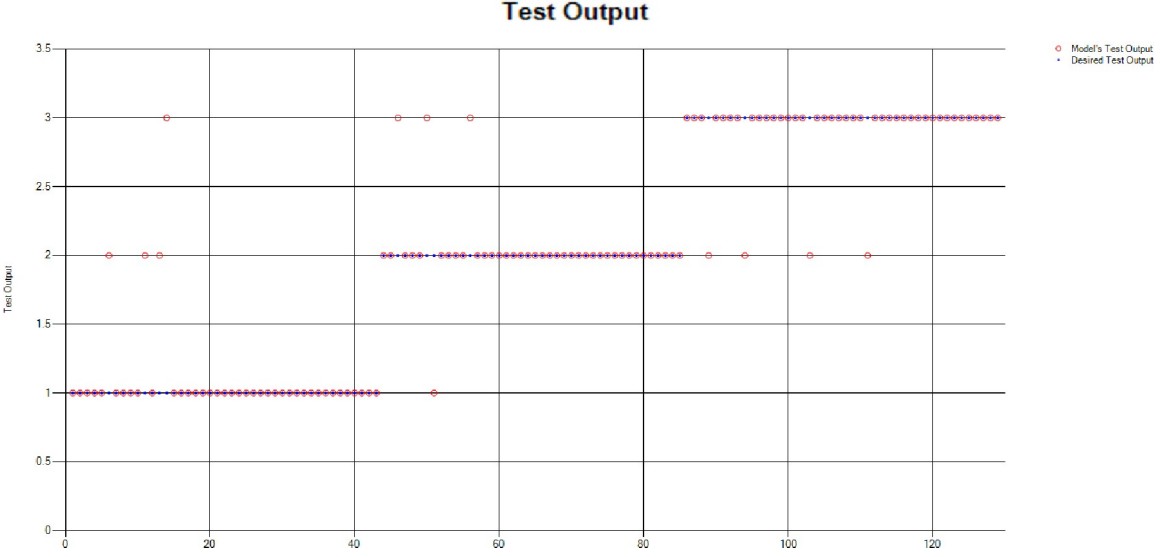

**Fig 28. Scenario 2 NB test outcome.**

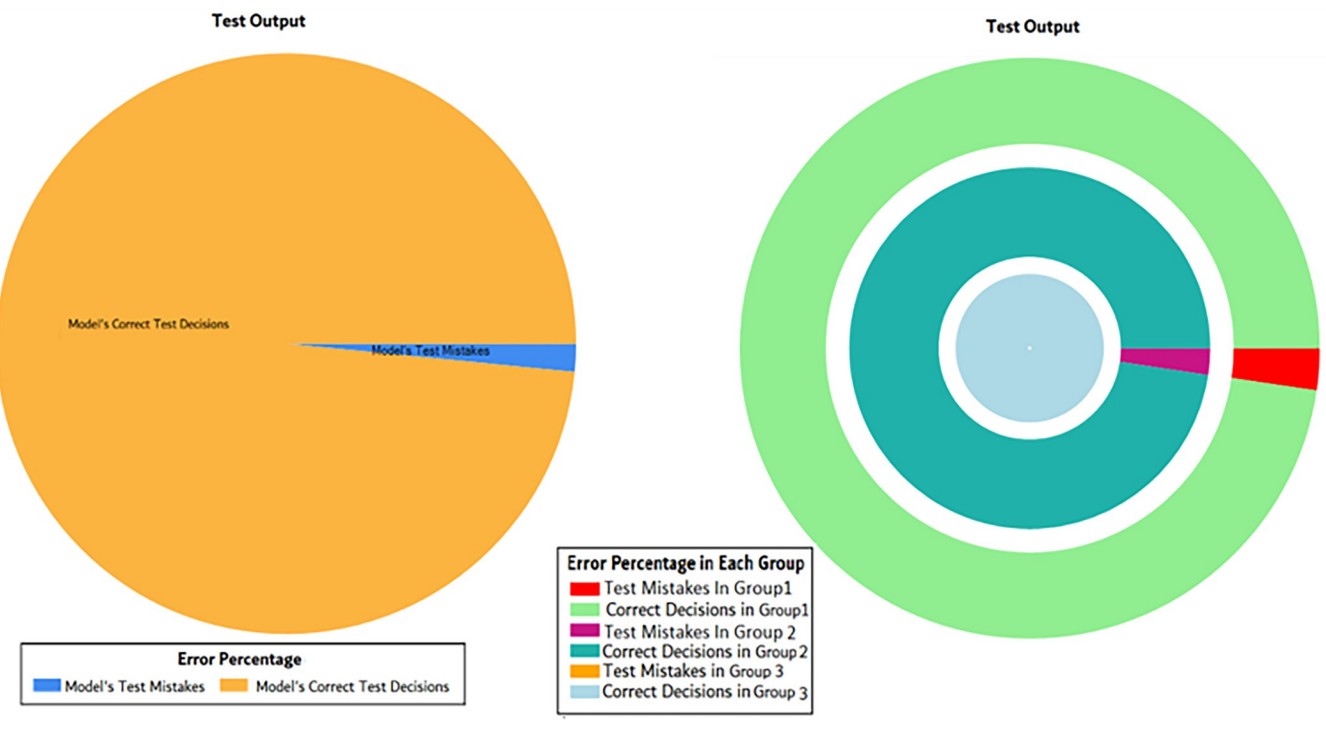

**Fig 29. Scenario 2 NB test overall accuracy percentage and distribution each group.**

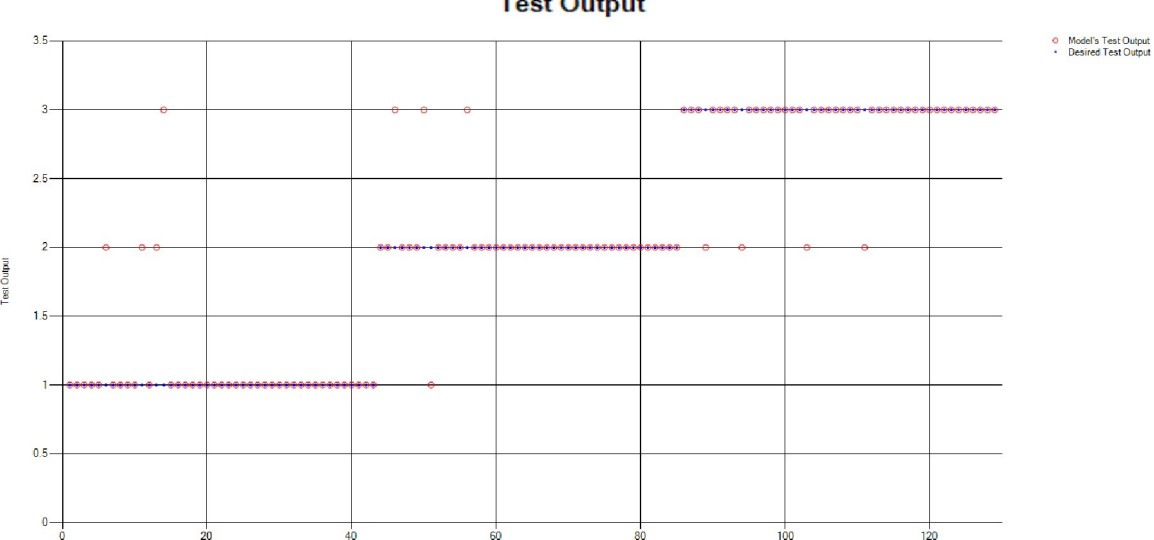

**Fig 30. Scenario 2 LR test outcome.**

The test outcome for LR for Scenario2 is shown in Fig 30. LR has had the lowest performance accuracy-wise among all the other methods. This could be due to the complex nature of the patterns in the data.

The test accuracy and the accuracy in each group has been demonstrated in Fig 31A and 31B for Scenario2 for LR. The test accuracy for each group is not as well as the other methods either which could be due to the complex nature of the data.

All the applied machine learning methods have provided relatively good results. However, for Scenario1, the best test accuracy was provided by RF and for Scenario2, the best test accuracy was given by KNN. GBM, SVM and RF ranked next after the best performing methods with regards to test accuracies.

## 4. Discussion

### 4.1 Applying strategic portfolio management to improve organizational agility

According to PMI [3,26–30], Portfolio Management also referred to as Strategic Portfolio Management (SPM) [47] aims to achieve strategic objectives through projects, programs and operations. SPM is a means to support agility by selecting the right projects, maximizing resource allocation, measuring and evaluating strategic alignment and balance to maximize the use of available resources [5,48–50].

SPM is required to support engagement and maintain organizational competitiveness [30] to support organizational agility. SPM implementation requires a supportive environment and consideration of risks, strategic foresight, constraints and barriers together with other impacting factors for SPM implementation must be identified [5].

These may include, but not limited to, changes to work practices due to the pandemic, increased use of technology, law limitations, market conditions, government or industry standards, infrastructure, human resource management and organizational process assets. In addition to this it will include project management information systems, methods for knowledge sharing and agile project delivery approaches.

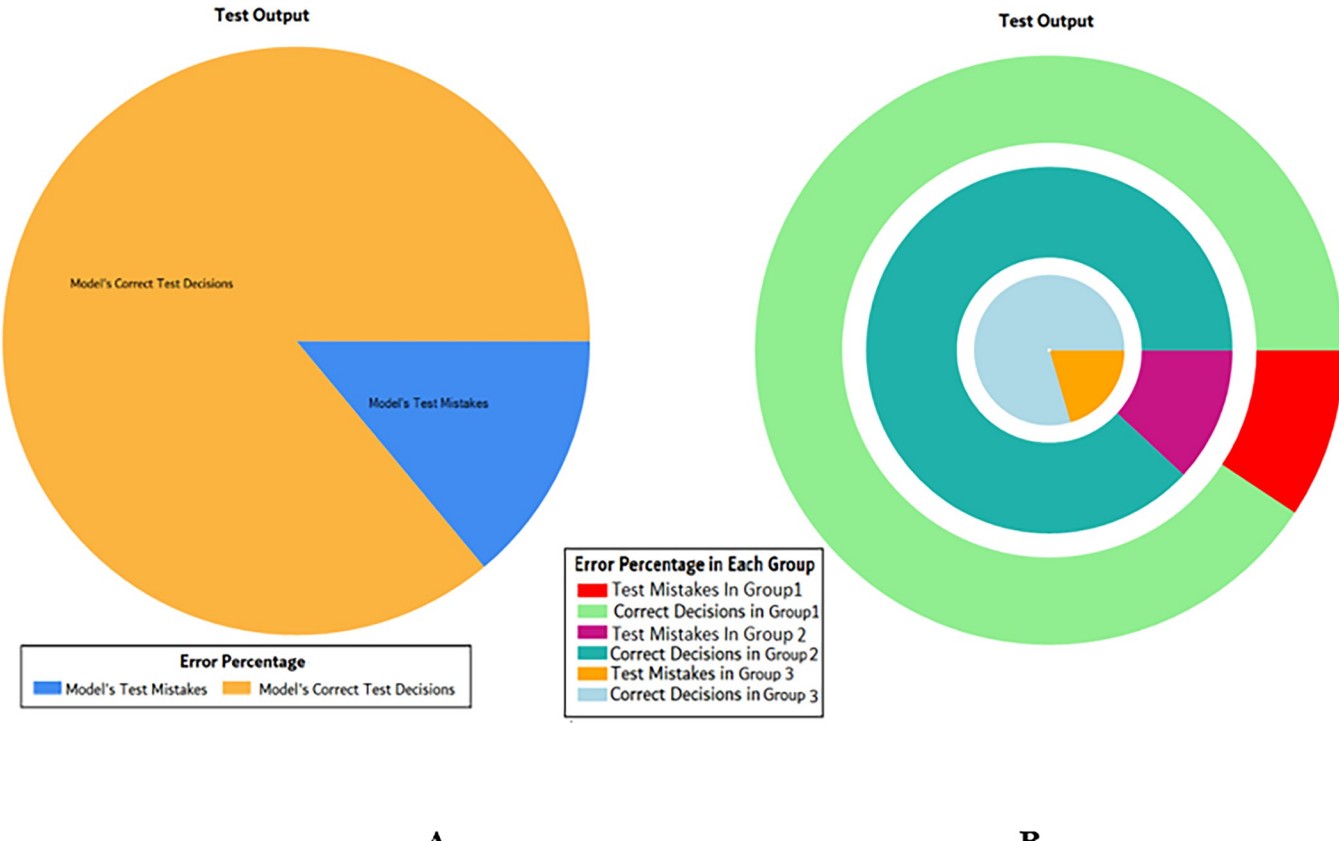

**Fig 31. Scenario 2 LR test overall accuracy percentage and distribution each group.**

Benefits of SPM include organizations becoming more flexible, dynamic, innovative, creative, communicative, strategically oriented, efficient and motivated [51]. Janssen [52] highlights the important lessons learnt for organisational agility, in particular post-pandemic. Incorporating organisational crisis/disaster management is one of the most important elements in bringing self-sustainability in businesses via keeping agility on the top of strategic vision [14,53,54]. Organisational agility and flexibility in decision marking [55–57] become more important to empower individuals, and businesses to innovate, satisfy the changing needs of the customers and swiftly adapt to business fluctuation [58–61].

Heising [9] views SPM as a support structure for organizational agility and a strategic role that enables organizations to respond and adapt to changing environmental conditions by monitoring and managing the project portfolio.

Killen & Hunt [48] stated that organizational agility supports strategic flexibility by enabling organisations to anticipate, identify, develop and implement change and investment strategies that are aligned with strategic objectives.

Hadjinicolaou et al [6] explored diverse characteristics of organizational agility and how diverse combinations existed within organisations at different levels of strategic management maturity.

Three sets of data containing 21 variables of Project Portfolio Management and organizational agility were analysed using canonical analysis. The results showed a significant positive strong correlation between SPM maturity and organizational agility characteristics within the studied groups. The research identified various combinations of characteristics constructed at

different levels of maturity, and specific characteristics that highly contribute to the highest levels of maturity [6].

## 4.2 Aspects of future change

In respect to aspects of future change respondents were asked about the degree of expected change in the next two years whether a Major Decrease, Minor Decrease, Stay about the same, Minor Increase or Major Increase of the following:

1. Revenue/Budget

2. Profit/Surplus

3. Market Share

4. Market Size for our products or services

5. Number of employees

6. Competition

7. Impacts of Legislative Changes

8. Impacts from Globalisation

9. Impacts from technology and digital disruption

10. Impacts from acquisitions, management restructures

11. A greater number of small projects

12. A greater number of large projects

The response to the survey questions is presented in Fig 32 regarding the aforementioned aspects, which represents that some parameters mostly remain the same e.g., the impact of globalisation, however, other parameters i.e., the competition and revenue/profits related changes are imminent.

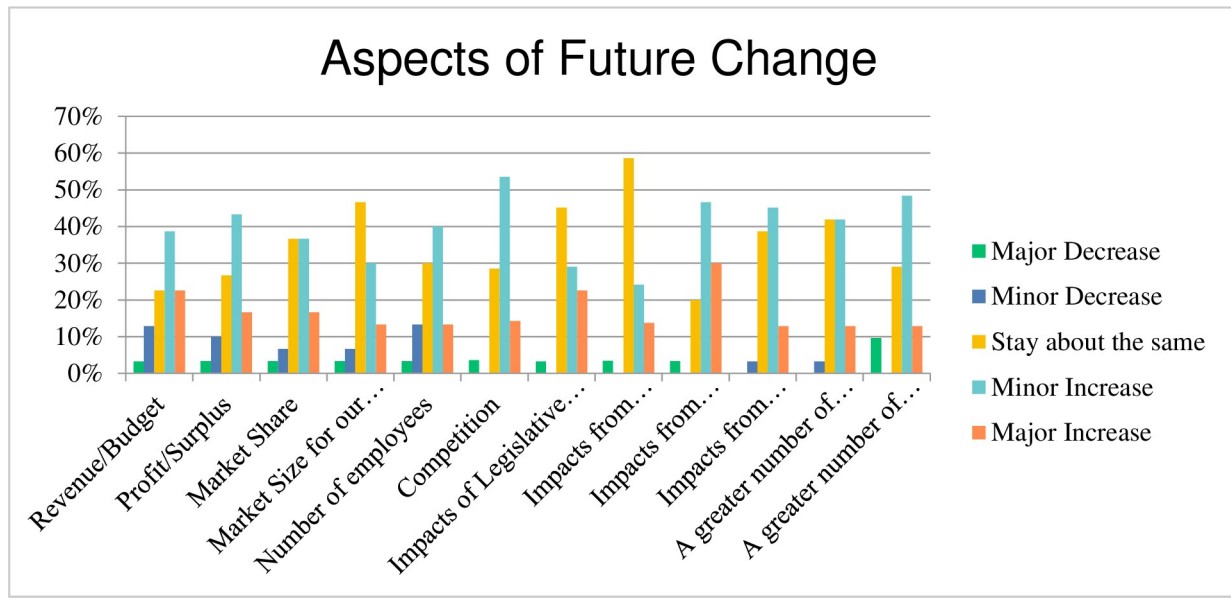

**Fig 32. Aspects of future change.**

### 4.3 Interpretation of results

**4.3.1 Benefits of organisational agility.** The results of this study found the top five benefits of organizational agility of statistical significance were:

1. More efficient ability to move from strategy formulation to execution

2. Quickly adapt practices based on learnings from successes and failures

3. Faster response to changing market conditions

4. Improved organisational efficiency overall

5. Improved customer satisfaction

These were in alignment with a previous study exploring the relationships between 20 agile characteristics and 15 perceived benefits which included improved responsiveness, efficiency, sustainability or business decision making. The top three perceived benefits for organizational agility found in that study were:

1) "Faster response to changing market conditions"; 2) "Greater cost-saving" and 3) "More effective targeting of innovation opportunities" [7].

Each of these are fundamental to improving organisational performance with a focus on innovation. These benefits also highlight the need for organizational agility and the benefits that it brings however attention is required to overcome the barriers of organizational agility.

**4.3.2 Barriers of organizational agility.** The top six barriers of Organizational Agility found in this study were:

1. Lack of executive engagement/leadership

2. Cultural mindset does not support agility

3. Lack of effective executive sponsorship

4. Lack of communication between departments

5. Slow decision making

6. Unclear organisational vision or strategy

Each of these involve action required at the executive leadership levels to improve engagement, focus on the culture, provide support and improved communication, reduce the time taken to make decisions and also define an organizations vision and strategy. A component of this will be the adoption of change management.

Change management practices adopted within organizations include:

1. Use of a PMO for change management

2. Work across organizational silos

3. Implement a formal change management process

4. Assign change managers

Each of these play an important part to improve organizational agility and implement practices required to improve maturity levels to support agility.

**4.3.3 Aspects of organisational maturity.** In line with the barriers for organizational agility respondents were asked about aspects of organizational maturity levels in the following areas:

1. Organizational process and procedure maturity

2. Product Development Maturity

3. Project Management Maturity

4. Project Portfolio Management Maturity

5. Organizational Resilience

6. Organizational Agility

7. Benefits Lifecycle Management

8. A focus on risk management

9. Value Management

The top five areas that received the lowest levels of maturity were:

1. Project Portfolio Management Maturity

2. Benefits Lifecycle Management

3. Organizational Agility

4. Product Development Maturity

5. Organizational process and procedure maturity

This indicates areas that work is required to improve these levels and practices to support future organizational agility to improve performance and deliver on strategy. Project Portfolio Management supports organizational agility as organisations develop agility practices and improve performance to adapt to external pressures [6].

Heising [9] views Project Portfolio Management as a strategic role that supports organizational agility by enabling organizations to respond and adapt to changing environmental conditions by monitoring and altering the project portfolio. Killen & Hunt [48] discovered connections between PPM and organizational agility in supporting organizations with responsive decision making and resource allocation as well as building dynamic capability at organizational level.

In terms of key areas of changes for an organisation, respondents reported external factors causing change were: 1) customers; 2) funding or funding sources; 3) Socio Political and 4) Legislation. These factors require adequate risk management and strategic foresight with comprehensive scanning and monitoring of external environments to ensure minimal impact on the organisation and associated development teams. The monitoring and scanning for external changes play a key role in the development of organizational agility.

## 5. Conclusion

The objective of this study was to examine the extent to which artificial intelligence could be used to predict an organization's future agility to mitigate risks and changes that may be required to support improving organisational agility. One of the limitations of the study was the insufficient sample size of the data collected requiring the Since the numbers of the respondents in the survey were not sufficient, firstly we have increased the sample size by producing synthetic data modelled on the data collected. After that the three efficient Machine Learning Models (Support Vector Machine (SVM), Decision Tree (DT) and k-Nearest Neighbours (KNN), Gradient Boosting Machine (GBM), Random Forest (RF), Naïve Bayes (NB) and

Logistic Regression (LR)) have been used to predict agility among the participants. The overall test accuracy for most of the models is 90+% where GBM, RF, SVM and KNN are ranked higher among the other methods. The best-performing model is RF for Scenario 1 with the test accuracy of 97.674% and KNN with the test accuracy of 99.224 in Scenario 2. This means that using the above-mentioned model, the proposed system would be able to predict the Organizational Agility using the responses from the survey questions. Considering the importance of using technology to enhance informed decision-making, Artificial Intelligence methods proposed in the current study have achieved an accurate prediction outcome and performed the prediction task efficiently.

Modelling was developed using organizational attributes, principles and practices that are being utilized by industry to develop a culture of agility as applied to an organizations projects and operational work. The research data was collected from 44 respondents in public and private Australian industry sectors with varying sizes and industry sectors. These research findings were used to develop an Artificial Intelligence model to predict and support future organizational agility. The model included predominate agile principles and practices, aspects of size and type of organization, support structures being adopted critical to success, maturity levels and aspects of future change.

The findings from the research data collected (which addresses objective 1 of the study) supported the requirement of organisations to focus on found amongst the leading characteristics such as of organizational agility were having open communications; being flexible and adaptable; having transparency in decision making with continuous learning from experience; self-awareness and honesty; empowering team members with openness to change and having a commitment to agility.

The leading practices from the research data collected for organizational agility were being responsive to strategic opportunities; eliminating organisational silos; focusing on change management; having shorter production/review/decision cycles; integrating the voice of the customer; using interdisciplinary project teams; leveraging technology and an ongoing assessment of disruptive technological or other changes. Many of these practices to support agility and crisis management require commitment to an ongoing cycle of people, process and technology changes together with a culture of continuous improvement and responsiveness to internal and external change.

The findings from the research data collected also identified challenges and barriers to improving organizational agility together with the benefits of improved organizational agility. The barriers included the lack of executive engagement and leadership; the cultural mindset not supporting agility; lack of effective executive sponsorship; lack of communication between departments; slow decision making and unclear organisational vision or strategy. All of which require attention and mitigation of risks that reduce agility. The benefits and reasons to focus on organisational agility included improving the ability to move from strategy formulation to execution more efficiently; quickly adapting practices based on learnings from successes and failures; a faster response to changing market conditions; improved organisational efficiency overall and improved customer satisfaction.

A further limitation of this study is the data collected was from Australian organisations and prior to the Covid-19 pandemic. A further and broader collection would further test the model and potentially uncover changes as a result of new ways of working and working from home in a future study.

As a result of this study and the application of machine learning to predict agility organizations are able to apply to identify gaps and apply the framework to build new skills and competencies required, apply new strategic practices and improving the culture through training and

education to focus required on innovation, change management and maturity levels to support engagement and continuous improvement.

The focus on cultural change and change management requires consideration of PMOs and other support structures, working across organizational silos, assignment of change champions and implementation of change processes to mitigate risks. Monitoring of internal and external changes also becomes critical to build agility to manage risks, apply strategic foresight, stakeholder engagement and deliver on strategy.

Our study makes a valuable contribution to emerging literature and an understanding of the principles, practices and characteristics that will help organizations with limited resources overcome barriers to build a framework and culture of agility to delivery on strategy. This will also support the building of organizational maturity levels for ongoing processes, product and project management, portfolio management, benefits lifecycle management, risk and value management.

Further work could be undertaken to apply the model to create a baseline within an organisation, apply aspects of change and resurvey after twelve months to analyse findings and create a case study.

## Author Contributions

**Conceptualization:** Niusha Shafiabady, Nick Hadjinicolaou.

**Data curation:** Niusha Shafiabady, Nick Hadjinicolaou.

**Formal analysis:** Niusha Shafiabady.

**Investigation:** Nick Hadjinicolaou.

**Methodology:** Niusha Shafiabady, James Vakilian.

**Project administration:** Niusha Shafiabady.

**Software:** Niusha Shafiabady, Binayak Bhandari.

**Supervision:** Niusha Shafiabady.

**Validation:** Niusha Shafiabady.

**Visualization:** Robert M. X. Wu, James Vakilian.

**Writing – original draft:** Nick Hadjinicolaou, Fareed Ud Din.

**Writing – review & editing:** Robert M. X. Wu, James Vakilian.

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
