## [Decision Letter · Decision Letter 0]

16 Nov 2022

PONE-D-22-29717Using Artificial Intelligence (AI) Techniques to Predict Organizational AgilityPLOS ONE

Dear Dr. Shafiabady,

Thank you for submitting your manuscript to PLOS ONE. After careful consideration, we feel that it has merit but does not fully meet PLOS ONE’s publication criteria as it currently stands. Therefore, we invite you to submit a revised version of the manuscript that addresses the points raised during the review process.

We look forward to receiving your revised manuscript.

Kind regards,

Sathishkumar V E

Academic Editor

PLOS ONE

Journal Requirements:

3. Please provide additional details regarding ethical approval in the body of your manuscript. In the Methods section, please ensure that you have specified the name of the IRB/ethics committee that approved your study.

"NO"

Reviewers' comments:

Reviewer's Responses to Questions

**Comments to the Author**

1. Is the manuscript technically sound, and do the data support the conclusions?

Reviewer #1: Partly

Reviewer #2: No

Reviewer #3: Yes

2. Has the statistical analysis been performed appropriately and rigorously? 

Reviewer #1: Yes

Reviewer #2: No

Reviewer #3: Yes

3. Have the authors made all data underlying the findings in their manuscript fully available?

Reviewer #1: No

Reviewer #2: No

Reviewer #3: Yes

4. Is the manuscript presented in an intelligible fashion and written in standard English?

Reviewer #1: Yes

Reviewer #2: Yes

Reviewer #3: Yes

5. Review Comments to the Author

Reviewer #1: Using Artificial Intelligence (AI) Techniques to Predict Organizational Agility

The topics of this paper are interesting. The structure and content must be revised, and results have to be better explained by authors before to be reconsidered for publication.

Abstract is messy. It has to be shorter. Authors have to clarify in the following order aim, methods, empirical results and implications of organizational management concerning organizational agility in context of crisis management, such as for pandemic crisis.

Introduction has to better clarify the research questions of this study and provide more theoretical background of these topics in context of crisis management to cope with pandemic threats. Long lists of items in introduction can be compressed in orizontal lines and not vertical ones (See suggested readings that must be all read and used in the text to reinforce theoretical background).

Authors writes of AI in title but they focus on section 2 on machine learning. They have to clarify if they focus on AI that is a large set or subsets of machine learning and deep learning 8see suggested paper to clarify the differences).

Methods of this study is messy and not clear. The section of Materials and methods must be re-structured with following three sections only and order:

• Sample and data

• Measures of variables

• Data analysis procedure.

Section 3. Research questions have to be put in introduction to clarify the aim. Moreover section 3 called research design that is similar to methodology is misleading. The section called as Results is OK.

Moreover, authors have to avoid subheadings that create fragmentation and confusion. If necessary, can use bullet points (same comments for all sections).

Figure 1. Title must be below and not above the figure.

Clarify unit of measure on y-axis (% of total?) The items on x-axis can be put in decreasing order from left to right and not confused manner.

Figure 2, same comments of figure 1.

Figure 3 same comments. Moreover 19 etc. are employees and has to be specified. This categorization has to be clarified in methods, indicating the sources authors focus on micro, small, medium, etc. enterprises?

Clarify.

Section 4 is about other results or is only discussion? Call the section properly if this is only "discussion".

To reiterate, avoiding in the just mentioned sections, sub-headings that create fragmentation of the paper.

Authors have the tendency to create long list of items when they can be compressed in lines such as 1)aa, 2)bb, 3cc) etc.…

Table 1. in title avoids acronyms or clarify it in a note to the table.

Groups have to be clarified always in a note. To describe them in the text is not enough. Three decimal number is enough.

Table 2 has to clarify acronyms in a note as explained before. Moreover, same comments of table 1.

Figure 4-15 the text within and legends are too small and cannot be read. Enlarge them.

Moreover, avoid acronyms in titles. The paper has a lot of figures and tables that are difficult to digest, some of them can be put in appendix and inserting in the text the most important ones to improve the readability…

The paper is too long and has to be reduced. Rewrite the long lists of items in lines.

The paper is messy and boring and has to be shortened.

To reiterate again, authors have to avoid subheadings that create fragmentation and confusion. If necessary, can use bullet points.

Conclusion has not to be a summary, but authors have to focus on manifold limitations of this study and provide suggestions of organizational and managerial behaviour in context of crisis management, clarifying the role of machine learning applied.

Overall, then, the paper is interesting, but the content is messy and vague and in some parts is inconsistent. Theoretical framework is weak and has to be reinforced with all suggested papers, and some results create confusion and have to be clarified… structure of the paper has to be improved; study design, discussion and presentation of results have to be clarified using suggested comments.

I strongly suggest revising considering and using strictly all comments (references included) that I will in-depth verify, to consider the paper in the journal and avoid rejection.

Suggested readings of relevant papers that have to be read and all inserted in the text and references.

Janssen, M.; van der Voort, H. Agile and adaptive governance in crisis response: Lessons from the COVID-19 pandemic. Int. J. Inf. Manag. 2020, 55, 102180

Coccia M. 2020. Deep learning technology for improving cancer care in society: New directions in cancer imaging driven by artificial intelligence. Technology in Society, vol. 60, February, pp. 1-11, art. n. 101198, https://doi.org/10.1016/j.techsoc.2019.101198

Solheim, M.C.W., Aadland, T., Eide, A.E., Haneberg, D.H. 2022Drivers for agility in times of crisis.European Business Review Article in Press

El Idrissi, M., El Manzani, Y., Ahl Maatalah, W., Lissaneddine, Z. 2022.Organizational crisis preparedness during the COVID-19 pandemic: an investigation of dynamic capabilities and organizational agility roles.International Journal of Organizational Analysis

Ardito L., Coccia M., Messeni Petruzzelli A. 2021. Technological exaptation and crisis management: Evidence from COVID-19 outbreaks. R&D Management, vol. 51, n. 4, pp. 381-392, https://doi.org/10.1111/radm.12455

Evans, S.; Bahrami, H. Super-Flexibility in Practice: Insights from a Crisis. Glob. J. Flex. Syst. Manag. 2020, 21, 207–214.

Coccia M. 2021. Comparative Critical Decisions in Management. In: Farazmand A. (eds), Global Encyclopedia of Public Administration, Public Policy, and Governance. Springer Nature, Cham. https://doi.org/10.1007/978-3-319-31816-5_3969-1

Crow, D.A.; Albright, E.A.; Ely, T.; Koebel, E.; Lawhon, L. Do disasters lead to learning? Financial policy change in local government. Rev. Policy Res. 2018, 35, 564–589

Coccia M. 2021e. Pandemic Prevention: Lessons from COVID-19. Encyclopedia, vol. 1, n. 2, pp. 433-444. doi: 10.3390/encyclopedia1020036

Mero, J., Haapio, H. 2022 An effectual approach to executing dynamic capabilities under unexpected uncertainty.Industrial Marketing Management107, pp. 82-91

Coccia M. 2019. Intrinsic and extrinsic incentives to support motivation and performance of public organizations, Journal of Economics Bibliography, vol. 6, no. 1, pp. 20-29, http://dx.doi.org/10.1453/jeb.v6i1.1795

Akkaya, B., Mert, G. 2022Organizational Agility, Competitive Capabilities, and the Performance of Health Care Organizations during the Covid-19 Pandemic. Central European Management Journal30(1), pp. 2-25

Anton, P. 2021Organisational agility in a context of crisis | Soins66(856), pp. 61-65

Reviewer #2: The manuscript objective is to use AI to predict organizational future agility.

The primary research objectives set are shortlisting the leading characteristics and foundational practices to support organizational agility? and how one can make use of AI to predict organizational agility. Its evident from literature that AI is preferred tool in several Industrial applications.

However, in the manuscript, the approach or case study is not presented, should be presented to readers in a very comprehensive manner. It would be much better to provide more detailed background information and quantitative indicators comparative analysis before and after the AI implementation.

The manuscript miss in presenting information related to AI model/flowchart and its implementation, results and discussion numerically and statistically. The contribution of this paper is not clear. The authors should at least compare with some of existing methodology/model to show the advantage of the proposed methodology here in context. In section 4, the organizational agility metrics/criteria/barriers presented are limited/repetitive and are mostly from past literature. What is contribution/addition of the study here? Only environmental and social factors, are they enough for organizational agility? Whatever top practices need for organizational agility studied or found are very generic type, how those are incorporated in AI model to estimate organizational agility is interesting and this is missing in the manuscript. The manuscript says, data is collected from a comprehensive 44 different industries, but what information is collected is missing, should be presented as a sample. The manuscript lags in presenting the approach adopted and its logical/mathematical analysis to estimate/predict organizational agility. Need to provide a rational and reasonable comparison with the pertinent models and adopted model. Need to present the outline of the manuscript in introduction. And add future research direction in conclusion. Need to reframe sections and their titles to maintain/improve flow and readability of manuscript. The manuscript written in standard English, but not presented in an intelligible fashion.

Thank you, wish you all best of luck.

Reviewer #3: What is the motivation of the proposed work? Research gaps, objectives of the proposed work should be clearly justified

The literature has to be strongly updated with some relevant and recent papers focused on the fields dealt with in the manuscript. Include references related to the study proposed as there are lot of references related to Machine learning algorithms.

Explain why the AI method was selected for the study, its importance and compare with traditional methods.

In section 2.1, Authors explained ML algorithms and given the title as AI methods. Whether AI or ML is used to predict the organisational aglity? Explain how AI and ML is related?

SVM, DT and KNN are used for prediction. Authors should consider algorithms such as Gradient boosting machine, Random forest, Naive bayes, logistic regresson, etc.

Other than Accuracy, Sensitivity, specificity, F1 score, recall also should be included.

What about hyperparameters selection? Explain.

Authors are suggested to include more discussion on the results and also include some explanation regarding the justification to support why the proposed method is better in comparison towards other methods

Does this kind of study have never attempted before? Justify this statement and give an appropriate explanation to do so in this paper.

Quality of figures is so important too. Please provide some high-resolution figures. Some figures have a poor resolution.

Change the title accordingly AI or ML based approach. Title should be more specific.

Include limitations of the study.

Overall presentation should be improved.

6. PLOS authors have the option to publish the peer review history of their article (what does this mean?). If published, this will include your full peer review and any attached files.

Reviewer #1: No

Reviewer #2: No

Reviewer #3: **Yes: **Usha Moorthy

---

## [Author Response · Author response to Decision Letter 0]

19 Feb 2023

I have attached a word copy of the response. In case it might not be formatted properly here. 

Table 1: Reviewers’ Responses

Item no Response

Reviewer1 We would like to thank Reviewer 1 for their excellent comments. We do appreciate your comments, they were very helpful. We have modified and restructured the paper accordingly and addressed your comments. Thank you for the references you mentioned which were very useful. We have enhanced the quality of the images, added the missing information to the tables, added and omitted the necessary and extra information. Strengthened the theoretical framework and addressed your comments although the manuscript in different sections. I have included the comments and the detailed response in Table 2 below.

Reviewer2 We would like to thank Reviewer 2 for their great comments. We have addressed the comments in detail within the updated version of the manuscript. 

Reviewer3 We would like to thank Reviewer 3 for their excellent comments. 

Table 2: Detailed Reviewers’ Responses

Item No 

Reviewer 1 Reviewer #1: Using Artificial Intelligence (AI) Techniques to Predict Organizational Agility

1) The topics of this paper are interesting. The structure and content must be revised, and results have to be better explained by authors before to be reconsidered for publication.

Response: We have revised the structure and the content of the manuscript.

2) Abstract is messy. It has to be shorter. Authors have to clarify in the following order aim, methods, empirical results and implications of organizational management concerning organizational agility in context of crisis management, such as for pandemic crisis.

Response: We have addressed this in revision. Added a section on pandemic in the paper.

3) Introduction has to better clarify the research questions of this study and provide more theoretical background of these topics in context of crisis management to cope with pandemic threats. Long lists of items in introduction can be compressed in orizontal lines and not vertical ones (See suggested readings that must be all read and used in the text to reinforce theoretical background).

Response: Have amended the paper and modified the paper accordingly. 

4) Authors writes of AI in title but they focus on section 2 on machine learning. They have to clarify if they focus on AI that is a large set or subsets of machine learning and deep learning 8see suggested paper to clarify the differences).

Response: Added a brief explanation in the section explain the difference and cited the reference.

5) Methods of this study is messy and not clear. The section of Materials and methods must be re-structured with following three sections only and order:

• Sample and data

• Measures of variables

• Data analysis procedure.

Response: Restructured the paper and changed the headings related to this section.

6) Section 3. Research questions have to be put in introduction to clarify the aim. Moreover section 3 called research design that is similar to methodology is misleading. The section called as Results is OK.

Moreover, authors have to avoid subheadings that create fragmentation and confusion. If necessary, can use bullet points (same comments for all sections).

Response: Restructured the paper and addressed this.

7) Figure 1. Title must be below and not above the figure.

Clarify unit of measure on y-axis (% of total?) The items on x-axis can be put in decreasing order from left to right and not confused manner.

Figure 2, same comments of figure 1.

Figure 3 same comments. Moreover 19 etc. are employees and has to be specified. 

Response: The issue about the Figures has been addresses. Also a paragraph is added explain the information which was missing.

8) This categorization has to be clarified in methods, indicating the sources authors focus on micro, small, medium, etc. enterprises?

Clarify.

Response: A paragraph is added explaining this.

9) Section 4 is about other results or is only discussion? Call the section properly if this is only "discussion".

Response: Changed it to Discussion.

10) To reiterate, avoiding in the just mentioned sections, sub-headings that create fragmentation of the paper.

Response: Done.

11) Authors have the tendency to create long list of items when they can be compressed in lines such as 1)aa, 2)bb, 3cc) etc.…

Response: Done.

12) Table 1. in title avoids acronyms or clarify it in a note to the table.

Response: Explanation added, thanks for mentioning it. 

13) Groups have to be clarified always in a note. To describe them in the text is not enough. Three decimal number is enough.

Response: Done.

14) Table 2 has to clarify acronyms in a note as explained before. Moreover, same comments of table 1.

Response: Explanation added. 

15) Figure 4-15 the text within and legends are too small and cannot be read. Enlarge them.

Response: I have redrawn all the figures to make them legible and enhance the quality.

16) Moreover, avoid acronyms in titles. The paper has a lot of figures and tables that are difficult to digest, some of them can be put in appendix and inserting in the text the most important ones to improve the readability…

Response: I have added the explanation for all the tables with acronyms.

17) The paper is too long and has to be reduced. Rewrite the long lists of items in lines.

The paper is messy and boring and has to be shortened.

To reiterate again, authors have to avoid subheadings that create fragmentation and confusion. If necessary, can use bullet points.

Response: Tried to reduce the subheading and restructured the paper. 

18) Conclusion has not to be a summary, but authors have to focus on manifold limitations of this study and provide suggestions of organizational and managerial behaviour in context of crisis management, clarifying the role of machine learning applied.

Response: Done.

19) Overall, then, the paper is interesting, but the content is messy and vague and in some parts is inconsistent. Theoretical framework is weak and has to be reinforced with all suggested papers, and some results create confusion and have to be clarified… structure of the paper has to be improved; study design, discussion and presentation of results have to be clarified using suggested comments.

Response: Done.

20) I strongly suggest revising considering and using strictly all comments (references included) that I will in-depth verify, to consider the paper in the journal and avoid rejection.

Suggested readings of relevant papers that have to be read and all inserted in the text and references.

Janssen, M.; van der Voort, H. Agile and adaptive governance in crisis response: Lessons from the COVID-19 pandemic. Int. J. Inf. Manag. 2020, 55, 102180

Coccia M. 2020. Deep learning technology for improving cancer care in society: New directions in cancer imaging driven by artificial intelligence. Technology in Society, vol. 60, February, pp. 1-11, art. n. 101198, https://doi.org/10.1016/j.techsoc.2019.101198

Solheim, M.C.W., Aadland, T., Eide, A.E., Haneberg, D.H. 2022Drivers for agility in times of crisis.European Business Review Article in Press

El Idrissi, M., El Manzani, Y., Ahl Maatalah, W., Lissaneddine, Z. 2022.Organizational crisis preparedness during the COVID-19 pandemic: an investigation of dynamic capabilities and organizational agility roles.International Journal of Organizational Analysis

Ardito L., Coccia M., Messeni Petruzzelli A. 2021. Technological exaptation and crisis management: Evidence from COVID-19 outbreaks. R&D Management, vol. 51, n. 4, pp. 381-392, https://doi.org/10.1111/radm.12455

Evans, S.; Bahrami, H. Super-Flexibility in Practice: Insights from a Crisis. Glob. J. Flex. Syst. Manag. 2020, 21, 207–214.

Coccia M. 2021. Comparative Critical Decisions in Management. In: Farazmand A. (eds), Global Encyclopedia of Public Administration, Public Policy, and Governance. Springer Nature, Cham. https://doi.org/10.1007/978-3-319-31816-5_3969-1

Crow, D.A.; Albright, E.A.; Ely, T.; Koebel, E.; Lawhon, L. Do disasters lead to learning? Financial policy change in local government. Rev. Policy Res. 2018, 35, 564–589

Coccia M. 2021e. Pandemic Prevention: Lessons from COVID-19. Encyclopedia, vol. 1, n. 2, pp. 433-444. doi: 10.3390/encyclopedia1020036

Mero, J., Haapio, H. 2022 An effectual approach to executing dynamic capabilities under unexpected uncertainty.Industrial Marketing Management107, pp. 82-91

Coccia M. 2019. Intrinsic and extrinsic incentives to support motivation and performance of public organizations, Journal of Economics Bibliography, vol. 6, no. 1, pp. 20-29, http://dx.doi.org/10.1453/jeb.v6i1.1795

Akkaya, B., Mert, G. 2022Organizational Agility, Competitive Capabilities, and the Performance of Health Care Organizations during the Covid-19 Pandemic. Central European Management Journal30(1), pp. 2-25

Anton, P. 2021Organisational agility in a context of crisis | Soins66(856), pp. 61-65

Response: Thank you, we have addressed the above and cited the proposed references.

Reviewer 2 Reviewer #2: The manuscript objective is to use AI to predict organizational future agility.

1) The primary research objectives set are shortlisting the leading characteristics and foundational practices to support organizational agility? and how one can make use of AI to predict organizational agility. Its evident from literature that AI is preferred tool in several Industrial applications.

However, in the manuscript, the approach or case study is not presented, should be presented to readers in a very comprehensive manner. It would be much better to provide more detailed background information and quantitative indicators comparative analysis before and after the AI implementation.

The manuscript miss in presenting information related to AI model/flowchart and its implementation, results and discussion numerically and statistically. The contribution of this paper is not clear. The authors should at least compare with some of existing methodology/model to show the advantage of the proposed methodology here in context. 

Response: We have amended the paper according to the recommendation and restructured the paper and added elaborations.

 2) In section 4, the organizational agility metrics/criteria/barriers presented are limited/repetitive and are mostly from past literature. What is contribution/addition of the study here? Only environmental and social factors, are they enough for organizational agility? Whatever top practices need for organizational agility studied or found are very generic type, how those are incorporated in AI model to estimate organizational agility is interesting and this is missing in the manuscript. The manuscript says, data is collected from a comprehensive 44 different industries, but what information is collected is missing, should be presented as a sample. 

Response: Thank you for the comment. We have added the missing information to the paper in accordance to your comments. 

3) The manuscript lags in presenting the approach adopted and its logical/mathematical analysis to estimate/predict organizational agility. Need to provide a rational and reasonable comparison with the pertinent models and adopted model. Need to present the outline of the manuscript in introduction. And add future research direction in conclusion. 

Response: We have modified the manuscript and added a section in the conclusion to address the issue.

4) Need to reframe sections and their titles to maintain/improve flow and readability of manuscript. The manuscript written in standard English, but not presented in an intelligible fashion.

Thank you, wish you all best of luck. 

Response: Thank you, we have addressed the comment and modified the paper accordingly.

Reviewer 3 Reviewer #3: 

1) What is the motivation of the proposed work? Research gaps, objectives of the proposed work should be clearly justified

Response: We have added a paragraph the justification.

2)The literature has to be strongly updated with some relevant and recent papers focused on the fields dealt with in the manuscript. Include references related to the study proposed as there are lot of references related to Machine learning algorithms. 

Response: We have updated the literature review and cited newly published relevant paper (13 more papers). 

Explain why the AI method was selected for the study, its importance and compare with traditional methods. 

Response: We have added this information to the methods.

In section 2.1, Authors explained ML algorithms and given the title as AI methods. Whether AI or ML is used to predict the organisational aglity? Explain how AI and ML is related? 

Response: We have addressed the comment and added the information.

SVM, DT and KNN are used for prediction. Authors should consider algorithms such as Gradient boosting machine, Random forest, Naive bayes, logistic regresson, etc. 

Response: Added all the above-mentioned methods with their respective results. It really made the paper more solid, thank you.

Other than Accuracy, Sensitivity, specificity, F1 score, recall also should be included. 

Response: Added as required by the comment.

What about hyperparameters selection? Explain. 

Response: Added the information in the manuscrtipt.

Authors are suggested to include more discussion on the results and also include some explanation regarding the justification to support why the proposed method is better in comparison towards other methods 

Response: Done.

Does this kind of study have never attempted before? Justify this statement and give an appropriate explanation to do so in this paper. 

Response: Added the explanation.

Quality of figures is so important too. Please provide some high-resolution figures. Some figures have a poor resolution. 

Response: Redrew the figures with better resolutions. Thanks for this.

Change the title accordingly AI or ML based approach. Title should be more specific. 

Response: It is basically ML but the reason we used AI is because the business experts who might be interested in reading the paper wouldn’t be very familiar to ML (everyone is more familiar with the term AI rather than ML), so that is why we used 

Include limitations of the study. 

Response: Done, added the limitations.

Overall presentation should be improved. 

Response: Restructured the paper and made it organised.

---

## [Editor Report · Decision Letter 1]

2 Mar 2023

Using Artificial Intelligence (AI) to Predict Organizational Agility

PONE-D-22-29717R1

Dear Dr. Shafiabady,

We’re pleased to inform you that your manuscript has been judged scientifically suitable for publication and will be formally accepted for publication once it meets all outstanding technical requirements.

Kind regards,

Sathishkumar V E

Academic Editor

PLOS ONE
---

## [Editor Report · Acceptance letter]

6 Mar 2023

PONE-D-22-29717R1 

Using Artificial Intelligence (AI) to Predict Organizational Agility 

Dear Dr. Shafiabady:

I'm pleased to inform you that your manuscript has been deemed suitable for publication in PLOS ONE. Congratulations! Your manuscript is now with our production department. 

Kind regards, 

on behalf of

Dr. Sathishkumar V E 

Academic Editor

PLOS ONE